# The promises and pitfalls of Stochastic Gradient Langevin Dynamics

**Nicolas Brosse, Éric Moulines**
Centre de Mathématiques Appliquées, UMR 7641,
Ecole Polytechnique, Palaiseau, France.
`nicolas.brosse@polytechnique.edu, eric.moulines@polytechnique.edu`

**Alain Durmus**
Ecole Normale Supérieure CMLA,
61 Av. du Président Wilson 94235 Cachan Cedex, France.
`alain.durmus@cmla.ens-cachan.fr`

## Abstract

Stochastic Gradient Langevin Dynamics (SGLD) has emerged as a key MCMC algorithm for Bayesian learning from large scale datasets. While SGLD with decreasing step sizes converges weakly to the posterior distribution, the algorithm is often used with a constant step size in practice and has demonstrated successes in machine learning tasks. The current practice is to set the step size inversely proportional to $N$ where $N$ is the number of training samples. As $N$ becomes large, we show that the SGLD algorithm has an invariant probability measure which significantly departs from the target posterior and behaves like Stochastic Gradient Descent (SGD). This difference is inherently due to the high variance of the stochastic gradients. Several strategies have been suggested to reduce this effect; among them, SGLD Fixed Point (SGLDFP) uses carefully designed control variates to reduce the variance of the stochastic gradients. We show that SGLDFP gives approximate samples from the posterior distribution, with an accuracy comparable to the Langevin Monte Carlo (LMC) algorithm for a computational cost sublinear in the number of data points. We provide a detailed analysis of the Wasserstein distances between LMC, SGLD, SGLDFP and SGD and explicit expressions of the means and covariance matrices of their invariant distributions. Our findings are supported by limited numerical experiments.

## 1 Introduction

Most MCMC algorithms have not been designed to process huge sample sizes, a typical setting in machine learning. As a result, many classical MCMC methods fail in this context, because the mixing time becomes prohibitively long and the cost per iteration increases proportionally to the number of training samples $N$. The computational cost in standard Metropolis-Hastings algorithm comes from 1) the computation of the proposals, 2) the acceptance/rejection step. Several approaches to solve these issues have been recently proposed in machine learning and computational statistics.

Among them, the stochastic gradient langevin dynamics (SGLD) algorithm, introduced in [33], is a popular choice. This method is based on the Langevin Monte Carlo (LMC) algorithm proposed in [16, 17]. Standard versions of LMC require to compute the gradient of the log-posterior at the current fit of the parameter, but avoid the accept/reject step. The LMC algorithm is a discretization of a continuous-time process, the overdamped Langevin diffusion, which leaves invariant the target distribution $\pi$. To further reduce the computational cost, SGLD uses unbiased estimators of the

gradient of the log-posterior based on subsampling. This method has triggered a huge number of works among others [1, 21, 2, 6, 8, 12, 24, 13, 4] and have been successfully applied to a range of state of the art machine learning problems [27, 23].

The properties of SGLD with decreasing step sizes have been studied in [31]. The two key findings in this work are that 1) the SGLD algorithm converges weakly to the target distribution $\pi$, 2) the optimal rate of convergence to equilibrium scales as $n^{-1/3}$ where $n$ is the number of iterations, see [31, Section 5]. However, in most of the applications, constant rather than decreasing step sizes are used, see [1, 8, 18, 22, 30, 32]. A natural question for the practical design of SGLD is the choice of the minibatch size. This size controls on the one hand the computational complexity of the algorithm per iteration and on the other hand the variance of the gradient estimator. Non-asymptotic bounds in Wasserstein distance between the marginal distribution of the SGLD iterates and the target distribution $\pi$ have been established in [10, 11]. These results highlight the cost of using stochastic gradients and show that, for a given precision $\epsilon$ in Wasserstein distance, the computational cost of the plain SGLD algorithm does not improve over the LMC algorithm; Nagapetyan et al. [25] reports also similar results on the mean square error.

It has been suggested to use control variates to reduce the high variance of the stochastic gradients. For strongly log-concave models, Nagapetyan et al. [25], Baker et al. [3] use the mode of the posterior distribution as a reference point and introduce the SGLDFP (Stochastic Gradient Langevin Dynamics Fixed Point) algorithm. Nagapetyan et al. [25], Baker et al. [3] provide upper bounds on the mean square error and the Wasserstein distance between the marginal distribution of the iterates of SGLDFP and the posterior distribution. In addition, Nagapetyan et al. [25], Baker et al. [3] show that the overall cost remains sublinear in the number of individual data points, up to a preprocessing step. Other control variates methodologies are provided for non-concave models in the form of SAGA-Langevin Dynamics and SVRG-Langevin Dynamics [13, 7], albeit a detailed analysis in Wasserstein distance of these algorithms is only available for strongly log-concave models [5].

In this paper, we provide further insights on the links between SGLD, SGLDFP, LMC and SGD (Stochastic Gradient Descent). In our analysis, the algorithms are used with a constant step size and the parameters are set to the standard values used in practice [1, 8, 18, 22, 30, 32]. The LMC, SGLD and SGLDFP algorithms define homogeneous Markov chains, each of which admits a unique stationary distribution used as a hopefully close proxy of $\pi$. The main contribution of this paper is to show that, while the invariant distributions of LMC and SGLDFP become closer to $\pi$ as the number of data points increases, on the opposite, the invariant measure of SGLD never comes close to the target distribution $\pi$ and is in fact very similar to the invariant measure of SGD.

In Section 3.1, we give an upper bound in Wasserstein distance of order 2 between the marginal distribution of the iterates of LMC and the Langevin diffusion, SGLDFP and LMC, and SGLD and SGD. We provide a lower bound on the Wasserstein distance between the marginal distribution of the iterates of SGLDFP and SGLD. In Section 3.2, we give a comparison of the means and covariance matrices of the invariant distributions of LMC, SGLDFP and SGLD with those of the target distribution $\pi$. Our claims are supported by numerical experiments in Section 4.

## 2   Preliminaries

Denote by $\mathbf{z} = \{z_i\}_{i=1}^N$ the observations. We are interested in situations where the target distribution $\pi$ arises as the posterior in a Bayesian inference problem with prior density $\pi_0(\theta)$ and a large number $N \gg 1$ of i.i.d. observations $z_i$ with likelihoods $p(z_i|\theta)$. In this case, $\pi(\theta) = \pi_0(\theta) \prod_{i=1}^N p(z_i|\theta)$. We denote $U_i(\theta) = -\log(p(z_i|\theta))$ for $i \in \{1, \ldots, N\}$, $U_0(\theta) = -\log(\pi_0(\theta))$, $U = \sum_{i=0}^N U_i$.

Under mild conditions, $\pi$ is the unique invariant probability measure of the Langevin Stochastic Differential Equation (SDE):

$$\mathrm{d}\theta_t = -\nabla U(\theta_t)\mathrm{d}t + \sqrt{2}\mathrm{d}B_t \ , \tag{1}$$

where $(B_t)_{t\geq 0}$ is a $d$-dimensional Brownian motion. Based on this observation, Langevin Monte Carlo (LMC) is an MCMC algorithm that enables to sample (approximately) from $\pi$ using an Euler discretization of the Langevin SDE:

$$\theta_{k+1} = \theta_k - \gamma \nabla U(\theta_k) + \sqrt{2\gamma} Z_{k+1} \ , \tag{2}$$

where $\gamma > 0$ is a constant step size and $(Z_k)_{k \geq 1}$ is a sequence of i.i.d. standard $d$-dimensional Gaussian vectors. Discovered and popularised in the seminal works [16, 17, 29], LMC has recently received renewed attention [9, 15, 14, 11]. However, the cost of one iteration is $Nd$ which is prohibitively large for massive datasets. In order to scale up to the big data setting, Welling and Teh [33] suggested to replace $\nabla U$ with an unbiased estimate $\nabla U_0 + (N/p) \sum_{i \in S} \nabla U_i$ where $S$ is a minibatch of $\{1, \ldots, N\}$ with replacement of size $p$. A single update of SGLD is then given for $k \in \mathbb{N}$ by

$$\theta_{k+1} = \theta_k - \gamma \left( \nabla U_0(\theta_k) + \frac{N}{p} \sum_{i \in S_{k+1}} \nabla U_i(\theta_k) \right) + \sqrt{2\gamma} Z_{k+1} . \qquad (3)$$

The idea of using only a fraction of data points to compute an unbiased estimate of the gradient at each iteration comes from Stochastic Gradient Descent (SGD) which is a popular algorithm to minimize the potential $U$. SGD is very similar to SGLD because it is characterised by the same recursion as SGLD but without Gaussian noise:

$$\theta_{k+1} = \theta_k - \gamma \left( \nabla U_0(\theta_k) + \frac{N}{p} \sum_{i \in S_{k+1}} \nabla U_i(\theta_k) \right) . \qquad (4)$$

Assuming for simplicity that $U$ has a minimizer $\theta^\star$, we can define a control variates version of SGLD, SGLDFP, see [13, 7], given for $k \in \mathbb{N}$ by

$$\theta_{k+1} = \theta_k - \gamma \left( \nabla U_0(\theta_k) - \nabla U_0(\theta^\star) + \frac{N}{p} \sum_{i \in S_{k+1}} \{\nabla U_i(\theta_k) - \nabla U_i(\theta^\star)\} \right) + \sqrt{2\gamma} Z_{k+1} . \qquad (5)$$

It is worth mentioning that the objectives of the different algorithms presented so far are distinct. On the one hand, LMC, SGLD and SGDLFP are MCMC methods used to obtain approximate samples from the posterior distribution $\pi$. On the other hand, SGD is a stochastic optimization algorithm used to find an estimate of the mode $\theta^\star$ of the posterior distribution. In this paper, we focus on the fixed step-size SGLD algorithm and assess its ability to reliably sample from $\pi$. For that purpose and to quantify precisely the relation between LMC, SGLD, SGDFP and SGD, we make for simplicity the following assumptions on $U$.

**H1.** *For all $i \in \{0, \ldots, N\}$, $U_i$ is four times continuously differentiable and for all $j \in \{2, 3, 4\}$, $\sup_{\theta \in \mathbb{R}^d} \left\| \mathrm{D}^j U_i(\theta) \right\| \leq \tilde{L}$. In particular for all $i \in \{0, \ldots, N\}$, $U_i$ is $\tilde{L}$-gradient Lipschitz, i.e. for all $\theta_1, \theta_2 \in \mathbb{R}^d$, $\|\nabla U_i(\theta_1) - \nabla U_i(\theta_2)\| \leq \tilde{L} \|\theta_1 - \theta_2\|$.*

**H2.** *$U$ is $m$-strongly convex, i.e. for all $\theta_1, \theta_2 \in \mathbb{R}^d$, $\langle \nabla U(\theta_1) - \nabla U(\theta_2), \theta_1 - \theta_2 \rangle \geq m \|\theta_1 - \theta_2\|^2$.*

**H3.** *For all $i \in \{0, \ldots, N\}$, $U_i$ is convex.*

Note that under **H**1, $U$ is four times continuously differentiable and for $j \in \{2, 3, 4\}$, $\sup_{\theta \in \mathbb{R}^d} \left\| \mathrm{D}^j U(\theta) \right\| \leq L$, with $L = (N + 1)\tilde{L}$ and where $\left\| \mathrm{D}^j U(\theta) \right\| = \sup_{\|u_1\| \leq 1, \ldots, \|u_j\| \leq 1} \mathrm{D}^j U(\theta)[u_1, \ldots, u_j]$. In particular, $U$ is $L$-gradient Lipschitz. Furthermore, under **H2**, $U$ has a unique minimizer $\theta^\star$. In this paper, we focus on the asymptotic $N \to +\infty$,. We assume that $\liminf_{N \to +\infty} N^{-1}m > 0$, which is a common assumption for the analysis of SGLD and SGLDFP [3, 5]. In practice [1, 8, 18, 22, 30, 32], $\gamma$ is of order $1/N$ and we adopt this convention in this article.

For a practical implementation of SGLDFP, an estimator $\hat{\theta}$ of $\theta^\star$ is necessary. The theoretical analysis and the bounds remain unchanged if, instead of considering SGLDFP centered w.r.t. $\theta^\star$, we study SGLDFP centered w.r.t. $\hat{\theta}$ satisfying $\mathbb{E}[\|\hat{\theta} - \theta^\star\|^2] = O(1/N)$. Such an estimator $\hat{\theta}$ can be computed using for example SGD with decreasing step sizes, see [26, eq.(2.8)] and [3, Section 3.4], for a computational cost linear in $N$.

## 3 Results

### 3.1 Analysis in Wasserstein distance

Before presenting the results, some notations and elements of Markov chain theory have to be introduced. Denote by $\mathcal{P}_2(\mathbb{R}^d)$ the set of probability measures with finite second moment and by

$\mathcal{B}(\mathbb{R}^d)$ the Borel $\sigma$-algebra of $\mathbb{R}^d$. For $\lambda, \nu \in \mathcal{P}_2(\mathbb{R}^d)$, define the Wasserstein distance of order 2 by

$$\mathrm{W}_2(\lambda, \nu) = \inf_{\xi \in \Pi(\lambda,\nu)} \left( \int_{\mathbb{R}^d \times \mathbb{R}^d} \|\theta - \vartheta\|^2 \, \xi(\mathrm{d}\theta, \mathrm{d}\vartheta) \right)^{1/2} ,$$

where $\Pi(\lambda, \nu)$ is the set of probability measures $\xi$ on $\mathcal{B}(\mathbb{R}^d) \otimes \mathcal{B}(\mathbb{R}^d)$ satisfying for all $\mathsf{A} \in \mathcal{B}(\mathbb{R}^d)$, $\xi(\mathsf{A} \times \mathbb{R}^d)) = \lambda(\mathsf{A})$ and $\xi(\mathbb{R}^d \times \mathsf{A}) = \nu(\mathsf{A})$.

A Markov kernel $R$ on $\mathbb{R}^d \times \mathcal{B}(\mathbb{R}^d)$ is a mapping $R : \mathbb{R}^d \times \mathcal{B}(\mathbb{R}^d) \to [0, 1]$ satisfying the following conditions: (i) for every $\theta \in \mathbb{R}^d$, $R(\theta, \cdot) : \mathsf{A} \mapsto R(\theta, \mathsf{A})$ is a probability measure on $\mathcal{B}(\mathbb{R}^d)$ (ii) for every $\mathsf{A} \in \mathcal{B}(\mathbb{R}^d)$, $R(\cdot, A) : \theta \mapsto R(\theta, A)$ is a measurable function. For any probability measure $\lambda$ on $\mathcal{B}(\mathbb{R}^d)$, we define $\lambda R$ for all $\mathsf{A} \in \mathcal{B}(\mathbb{R}^d)$ by $\lambda R(\mathsf{A}) = \int_{\mathbb{R}^d} \lambda(\mathrm{d}\theta) R(\theta, \mathsf{A})$. For all $k \in \mathbb{N}^*$, we define the Markov kernel $R^k$ recursively by $R^1 = R$ and for all $\theta \in \mathbb{R}^d$ and $\mathsf{A} \in \mathcal{B}(\mathbb{R}^d)$, $R^{k+1}(\theta, \mathsf{A}) = \int_{\mathbb{R}^d} R^k(\theta, \mathrm{d}\vartheta) R(\vartheta, \mathsf{A})$. A probability measure $\bar{\pi}$ is invariant for $R$ if $\bar{\pi}R = \bar{\pi}$.

The LMC, SGLD, SGD and SGLDFP algorithms defined respectively by (2), (3), (4) and (5) are homogeneous Markov chains with Markov kernels denoted $R_{\mathrm{LMC}}, R_{\mathrm{SGLD}}, R_{\mathrm{SGD}}$, and $R_{\mathrm{FP}}$. To avoid overloading the notations, the dependence on $\gamma$ and $N$ is implicit.

**Lemma 1.** *Assume* **H1**, **H2** *and* **H3**. *For any step size* $\gamma \in (0, 2/L)$, $R_{\mathrm{SGLD}}$ *(respectively* $R_{\mathrm{LMC}}, R_{\mathrm{SGD}}, R_{\mathrm{FP}}$*) has a unique invariant measure* $\pi_{\mathrm{SGLD}} \in \mathcal{P}_2(\mathbb{R}^d)$ *(respectively* $\pi_{\mathrm{LMC}}, \pi_{\mathrm{SGD}}, \pi_{\mathrm{FP}}$*). In addition, for all* $\gamma \in (0, 1/L]$, $\theta \in \mathbb{R}^d$ *and* $k \in \mathbb{N}$,*

$$\mathrm{W}_2^2(R_{\mathrm{SGLD}}^k(\theta, \cdot), \pi_{\mathrm{SGLD}}) \leq (1 - m\gamma)^k \int_{\mathbb{R}^d} \|\theta - \vartheta\|^2 \, \pi_{\mathrm{SGLD}}(\mathrm{d}\vartheta)$$

*and the same inequality holds for LMC, SGD and SGLDFP.*

*Proof.* The proof is postponed to Section 1.1 in the supplementary document. $\qquad\square$

Under **H**1, (1) has a unique strong solution $(\theta_t)_{t \geq 0}$ for every initial condition $\theta_0 \in \mathbb{R}^d$ [20, Chapter 5, Theorems 2.5 and 2.9]. Denote by $(P_t)_{t \geq 0}$ the semigroup of the Langevin diffusion defined for all $\theta_0 \in \mathbb{R}^d$ and $\mathsf{A} \in \mathcal{B}(\mathbb{R}^d)$ by $P_t(\theta_0, \mathsf{A}) = \mathbb{P}(\theta_t \in \mathsf{A})$.

**Theorem 2.** *Assume* **H1**, **H2** *and* **H3**. *For all* $\gamma \in (0, 1/L]$, $\lambda, \mu \in \mathcal{P}_2(\mathbb{R}^d)$ *and* $n \in \mathbb{N}$, *we have the following upper-bounds in Wasserstein distance between*

    *i) LMC and SGLDFP,*

$$\mathrm{W}_2^2(\lambda R_{\mathrm{LMC}}^n, \mu R_{\mathrm{FP}}^n) \leq (1 - m\gamma)^n \, \mathrm{W}_2^2(\lambda, \mu) + \frac{2L^2\gamma d}{pm^2}$$
$$+ \frac{L^2\gamma^2}{p} n(1 - m\gamma)^{n-1} \int_{\mathbb{R}^d} \|\vartheta - \theta^\star\|^2 \, \mu(\mathrm{d}\vartheta) ,$$

    *ii) the Langevin diffusion and LMC,*

$$\mathrm{W}_2^2(\lambda R_{\mathrm{LMC}}^n, \mu P_{n\gamma}) \leq 2 \left( 1 - \frac{mL\gamma}{m+L} \right)^n \mathrm{W}_2^2(\lambda, \mu) + d\gamma \frac{m+L}{2m} \left( 3 + \frac{L}{m} \right) \left( \frac{13}{6} + \frac{L}{m} \right)$$
$$+ n\mathrm{e}^{-(m/2)\gamma(n-1)} L^3 \gamma^3 \left( 1 + \frac{m+L}{2m} \right) \int_{\mathbb{R}^d} \|\vartheta - \theta^\star\|^2 \, \mu(\mathrm{d}\vartheta) ,$$

    *iii) SGLD and SGD,*

$$\mathrm{W}_2^2(\lambda R_{\mathrm{SGLD}}^n, \mu R_{\mathrm{SGD}}^n) \leq (1 - m\gamma)^n \, \mathrm{W}_2^2(\lambda, \mu) + (2d)/m .$$

*Proof.* The proof is postponed to Section 1.2 in the supplementary document. $\qquad\square$

**Corollary 3.** *Assume* **H1**, **H2** *and* **H3**. *Set* $\gamma = \eta/N$ *with* $\eta \in (0, 1/(2\tilde{L})]$ *and assume that* $\liminf_{N \to \infty} mN^{-1} > 0$. *Then,*

*i) for all* $n \in \mathbb{N}$, *we get* $\mathrm{W}_2(R^n_{\mathrm{LMC}}(\theta^\star, \cdot), R^n_{\mathrm{FP}}(\theta^\star, \cdot)) = \sqrt{d\eta} \, O(N^{-1/2})$ *and* $\mathrm{W}_2(\pi_{\mathrm{LMC}}, \pi_{\mathrm{FP}}) = \sqrt{d\eta} \, O(N^{-1/2})$, $\mathrm{W}_2(\pi_{\mathrm{LMC}}, \pi) = \sqrt{d\eta} \, O(N^{-1/2})$.

*ii) for all* $n \in \mathbb{N}$, *we get* $\mathrm{W}_2(R^n_{\mathrm{SGLD}}(\theta^\star, \cdot), R^n_{\mathrm{SGD}}(\theta^\star, \cdot)) = \sqrt{d} \, O(N^{-1/2})$ *and* $\mathrm{W}_2(\pi_{\mathrm{SGLD}}, \pi_{\mathrm{SGD}}) = \sqrt{d} \, O(N^{-1/2})$.

Theorem 2 implies that the number of iterations necessary to obtain a sample $\varepsilon$-close from $\pi$ in Wasserstein distance is the same for LMC and SGLDFP. However for LMC, the cost of one iteration is $Nd$ which is larger than $pd$ the cost of one iteration for SGLDFP. In other words, to obtain an approximate sample from the target distribution at an accuracy $O(1/\sqrt{N})$ in 2-Wasserstein distance, LMC requires $\Theta(N)$ operations, in contrast with SGLDFP that needs only $\Theta(1)$ operations.

We show in the sequel that $\mathrm{W}_2(\pi_{\mathrm{FP}}, \pi_{\mathrm{SGLD}}) = \Omega(1)$ when $N \to +\infty$ in the case of a Bayesian linear regression, where for two sequences $(u_N)_{N\geq 1}$, $(v_N)_{N\geq 1}$, $u_N = \Omega(v_N)$ if $\liminf_{N\to+\infty} u_N/v_N > 0$. The dataset is $\mathbf{z} = \{(y_i, x_i)\}_{i=1}^N$ where $y_i \in \mathbb{R}$ is the response variable and $x_i \in \mathbb{R}^d$ are the covariates. Set $\mathbf{y} = (y_1, \dots, y_N) \in \mathbb{R}^N$ and $\mathbf{X} \in \mathbb{R}^{N \times d}$ the matrix of covariates such that the $i^{\mathrm{th}}$ row of $\mathbf{X}$ is $x_i$. Let $\sigma_y^2, \sigma_\theta^2 > 0$. For $i \in \{1, \dots, N\}$, the conditional distribution of $y_i$ given $x_i$ is Gaussian with mean $x_i^{\mathrm{T}}\theta$ and variance $\sigma_y^2$. The prior $\pi_0(\theta)$ is a normal distribution of mean $0$ and variance $\sigma_\theta^2 \, \mathrm{Id}$. The posterior distribution $\pi$ is then proportional to $\pi(\theta) \propto \exp\left(-(1/2)(\theta - \theta^\star)^{\mathrm{T}} \Sigma (\theta - \theta^\star)\right)$ where

$$\Sigma = \mathrm{Id}/\sigma_\theta^2 + \mathbf{X}^{\mathrm{T}}\mathbf{X}/\sigma_y^2 \quad \text{and} \quad \theta^\star = \Sigma^{-1}(\mathbf{X}^{\mathrm{T}}\mathbf{y})/\sigma_y^2 .$$

We assume that $\mathbf{X}^{\mathrm{T}}\mathbf{X} \succeq m \, \mathrm{Id}$, with $\liminf_{N\to+\infty} m/N > 0$. Let $S$ be a minibatch of $\{1, \dots, N\}$ with replacement of size $p$. Define

$$\nabla U_0(\theta) + (N/p) \sum_{i \in S} \nabla U_i(\theta) = \Sigma(\theta - \theta^\star) + \rho(S)(\theta - \theta^\star) + \xi(S)$$

where

$$\rho(S) = \frac{\mathrm{Id}}{\sigma_\theta^2} + \frac{N}{p\sigma_y^2}\sum_{i \in S} x_i x_i^{\mathrm{T}} - \Sigma , \quad \xi(S) = \frac{\theta^\star}{\sigma_\theta^2} + \frac{N}{p\sigma_y^2}\sum_{i \in S}\left(x_i^{\mathrm{T}}\theta^\star - y_i\right) x_i . \tag{6}$$

$\rho(S)(\theta - \theta^\star)$ is the multiplicative part of the noise in the stochastic gradient, and $\xi(S)$ the additive part that does not depend on $\theta$. The additive part of the stochastic gradient for SGLDFP disappears since

$$\nabla U_0(\theta) - \nabla U_0(\theta^\star) + (N/p)\sum_{i \in S}\left\{\nabla U_i(\theta) - \nabla U_i(\theta^\star)\right\} = \Sigma(\theta - \theta^\star) + \rho(S)(\theta - \theta^\star) .$$

In this setting, the following theorem shows that the Wasserstein distances between the marginal distribution of the iterates of SGLD and SGLDFP, and $\pi_{\mathrm{SGLD}}$ and $\pi$, is of order $\Omega(1)$ when $N \to +\infty$. This is in sharp contrast with the results of Corollary 3 where the Wasserstein distances tend to $0$ as $N \to +\infty$ at a rate $N^{-1/2}$. For simplicity, we state the result for $d = 1$.

**Theorem 4.** *Consider the case of the Bayesian linear regression in dimension* $1$.

*i) For all* $\gamma \in (0, \Sigma^{-1}\{1 + N/(p\sum_{i=1}^N x_i^2)\}^{-1}]$ *and* $n \in \mathbb{N}^*$,

$$\left(\frac{1 - \mu}{1 - \mu^n}\right)^{1/2} \mathrm{W}_2(R^n_{\mathrm{SGLD}}(\theta^\star, \cdot), R^n_{\mathrm{FP}}(\theta^\star, \cdot))$$

$$\geq \left\{2\gamma + \frac{\gamma^2 N}{p}\sum_{i=1}^N\left(\frac{(x_i\theta^\star - y_i)x_i}{\sigma_y^2} + \frac{\theta^\star}{N\sigma_\theta^2}\right)^2\right\}^{1/2} - \sqrt{2\gamma} ,$$

*where* $\mu \in (0, 1 - \gamma\Sigma]$.

*ii) Set* $\gamma = \eta/N$ *with* $\eta \in (0, \liminf_{N\to+\infty} N\Sigma^{-1}\{1 + N/(p\sum_{i=1}^N x_i^2)\}^{-1}]$ *and assume that* $\liminf_{N\to+\infty} N^{-1}\sum_{i=1}^N x_i^2 > 0$. *We have* $\mathrm{W}_2(\pi_{\mathrm{SGLD}}, \pi) = \Omega(1)$.

*Proof.* The proof is postponed to Section 1.3 in the supplementary document. $\square$

The study in Wasserstein distance emphasizes the different behaviors of the LMC, SGLDFP, SGLD and SGD algorithms. When $N \to \infty$ and $\lim_{N\to+\infty} m/N > 0$, the marginal distributions of the $k^{\text{th}}$ iterates of the LMC and SGLDFP algorithm are very close to the Langevin diffusion and their invariant probability measures $\pi_{\text{LMC}}$ and $\pi_{\text{FP}}$ are similar to the posterior distribution of interest $\pi$. In contrast, the marginal distributions of the $k^{\text{th}}$ iterates of SGLD and SGD are analogous and their invariant probability measures $\pi_{\text{SGLD}}$ and $\pi_{\text{SGD}}$ are very different from $\pi$ when $N \to +\infty$.

Note that to fix the asymptotic bias of SGLD, other strategies can be considered: choosing a step size $\gamma \propto N^{-\beta}$ where $\beta > 1$ and/or increasing the batch size $p \propto N^{\alpha}$ where $\alpha \in [0, 1]$. Using the Wasserstein (of order 2) bounds of SGLD w.r.t. the target distribution $\pi$, see e.g. [11, Theorem 3], $\alpha + \beta$ should be equal to 2 to guarantee the $\varepsilon$-accuracy in Wasserstein distance of SGLD for a cost proportional to $N$ (up to logarithmic terms), independently of the choice of $\alpha$ and $\beta$.

## 3.2 Mean and covariance matrix of $\pi_{\text{LMC}}, \pi_{\text{FP}}, \pi_{\text{SGLD}}$

We now establish an expansion of the mean and second moments of $\pi_{\text{LMC}}, \pi_{\text{FP}}, \pi_{\text{SGLD}}$ and $\pi_{\text{SGD}}$ as $N \to +\infty$, and compare them. We first give an expansion of the mean and second moments of $\pi$ as $N \to +\infty$.

**Proposition 5.** *Assume **H1** and **H2** and that $\liminf_{N\to+\infty} N^{-1}m > 0$. Then,*

$$\int_{\mathbb{R}^d} (\theta - \theta^\star)^{\otimes 2} \pi(\mathrm{d}\theta) = \nabla^2 U(\theta^\star)^{-1} + O_{N\to+\infty}(N^{-3/2}) \,,$$

$$\int_{\mathbb{R}^d} \theta \, \pi(\mathrm{d}\theta) - \theta^\star = -(1/2)\nabla^2 U(\theta^\star)^{-1} \, \mathrm{D}^3 \, U(\theta^\star)[\nabla^2 U(\theta^\star)^{-1}] + O_{N\to+\infty}(N^{-3/2}) \,.$$

*Proof.* The proof is postponed to Section 2.1 in the supplementary document. □

Contrary to the Bayesian linear regression where the covariance matrices can be explicitly computed, see Section 3 in the supplementary document, only approximate expressions are available in the general case. For that purpose, we consider two types of asymptotic. For LMC and SGLDFP, we assume that $\lim_{N\to+\infty} m/N > 0$, $\gamma = \eta/N$, for $\eta > 0$, and we develop an asymptotic when $N \to +\infty$. Combining Proposition 5 and Theorem 6 , we show that the biases and covariance matrices of $\pi_{\text{LMC}}$ and $\pi_{\text{FP}}$ are of order $\Theta(1/N)$ with remainder terms of the form $O(N^{-3/2})$, where for two sequences $(u_N)_{N\geq 1}, (v_N)_{N\geq 1}$, $u = \Theta(v)$ if $0 < \liminf_{N\to+\infty} u_N/v_N \leq \limsup_{N\to+\infty} u_N/v_N < +\infty$.

Regarding SGD and SGLD, we do not have such concentration properties when $N \to +\infty$ because of the high variance of the stochastic gradients. The biases and covariance matrices of SGLD and SGD are of order $\Theta(1)$ when $N \to +\infty$. To obtain approximate expressions of these quantities, we set $\gamma = \eta/N$ where $\eta > 0$ is the step size for the gradient descent over the normalized potential $U/N$. Assuming that $m$ is proportional to $N$ and $N \geq 1/\eta$, we show by combining Proposition 5 and Theorem 7 that the biases and covariance matrices of SGLD and SGD are of order $\Theta(\eta)$ with remainder terms of the form $O(\eta^{3/2})$ when $\eta \to 0$.

Before giving the results associated to $\pi_{\text{LMC}}, \pi_{\text{FP}}, \pi_{\text{SGLD}}$ and $\pi_{\text{SGD}}$, we need to introduce some notations. For any matrices $A_1, A_2 \in \mathbb{R}^{d\times d}$, we denote by $A_1 \otimes A_2$ the Kronecker product defined on $\mathbb{R}^{d\times d}$ by $A_1 \otimes A_2 : Q \mapsto A_1 Q A_2$ and $A^{\otimes 2} = A \otimes A$. Besides, for all $\theta_1 \in \mathbb{R}^d$ and $\theta_2 \in \mathbb{R}^d$, we denote by $\theta_1 \otimes \theta_2 \in \mathbb{R}^{d\times d}$ the tensor product of $\theta_1$ and $\theta_2$. For any matrix $A \in \mathbb{R}^{d\times d}$, $\mathrm{Tr}(A)$ is the trace of $A$.

Define K : $\mathbb{R}^{d\times d} \to \mathbb{R}^{d\times d}$ for all $A \in \mathbb{R}^{d\times d}$ by

$$\mathrm{K}(A) = \frac{N}{p} \sum_{i=1}^{N} \left( \nabla^2 U_i(\theta^\star) - \frac{1}{N}\sum_{j=1}^{N} \nabla^2 U_j(\theta^\star) \right)^{\otimes 2} A \,. \tag{7}$$

and H and G : $\mathbb{R}^{d\times d} \to \mathbb{R}^{d\times d}$ by

$$\mathrm{H} = \nabla^2 U(\theta^\star) \otimes \mathrm{Id} + \mathrm{Id} \otimes \nabla^2 U(\theta^\star) - \gamma \nabla^2 U(\theta^\star) \otimes \nabla^2 U(\theta^\star) \,, \tag{8}$$

$$\mathrm{G} = \nabla^2 U(\theta^\star) \otimes \mathrm{Id} + \mathrm{Id} \otimes \nabla^2 U(\theta^\star) - \gamma (\nabla^2 U(\theta^\star) \otimes \nabla^2 U(\theta^\star) + \mathrm{K}) \,. \tag{9}$$

K, H and G can be interpreted as perturbations of $\nabla^2 U(\theta^\star)^{\otimes 2}$ and $\nabla^2 U(\theta^\star)$, respectively, due to the noise of the stochastic gradients. It can be shown, see Section 2.2 in the supplementary document, that for $\gamma$ small enough, H and G are invertible.

**Theorem 6.** *Assume **H1**, **H2** and **H3**. Set $\gamma = \eta/N$ and assume that $\liminf_{N \to +\infty} N^{-1} m > 0$. There exists an (explicit) $\eta_0$ independent of $N$ such that for all $\eta \in (0, \eta_0)$,*

$$\int_{\mathbb{R}^d} (\theta - \theta^\star)^{\otimes 2} \pi_{\mathrm{LMC}}(\mathrm{d}\theta) = \mathrm{H}^{-1}(2\,\mathrm{Id}) + O_{N \to +\infty}(N^{-3/2})\,, \tag{10}$$

$$\int_{\mathbb{R}^d} (\theta - \theta^\star)^{\otimes 2} \pi_{\mathrm{FP}}(\mathrm{d}\theta) = \mathrm{G}^{-1}(2\,\mathrm{Id}) + O_{N \to +\infty}(N^{-3/2})\,, \tag{11}$$

*and*

$$\int_{\mathbb{R}^d} \theta \pi_{\mathrm{LMC}}(\mathrm{d}\theta) - \theta^\star = -\nabla^2 U(\theta^\star)^{-1} \mathrm{D}^3 U(\theta^\star)[\mathrm{H}^{-1}\,\mathrm{Id}] + O_{N \to +\infty}(N^{-3/2})\,,$$

$$\int_{\mathbb{R}^d} \theta \pi_{\mathrm{FP}}(\mathrm{d}\theta) - \theta^\star = -\nabla^2 U(\theta^\star)^{-1} \mathrm{D}^3 U(\theta^\star)[\mathrm{G}^{-1}\,\mathrm{Id}] + O_{N \to +\infty}(N^{-3/2})\,.$$

*Proof.* The proof is postponed to Section 2.2.2 in the supplementary document. $\square$

**Theorem 7.** *Assume **H1**, **H2** and **H3**. Set $\gamma = \eta/N$ and assume that $\liminf_{N \to +\infty} N^{-1} m > 0$. There exists an (explicit) $\eta_0$ independent of $N$ such that for all $\eta \in (0, \eta_0)$ and $N \geq 1/\eta$,*

$$\int_{\mathbb{R}^d} (\theta - \theta^\star)^{\otimes 2} \pi_{\mathrm{SGLD}}(\mathrm{d}\theta) = \mathrm{G}^{-1} \{2\,\mathrm{Id} + (\eta/p)\,\mathrm{M}\} + O_{\eta \to 0}(\eta^{3/2})\,, \tag{12}$$

$$\int_{\mathbb{R}^d} (\theta - \theta^\star)^{\otimes 2} \pi_{\mathrm{SGD}}(\mathrm{d}\theta) = (\eta/p)\,\mathrm{G}^{-1}\,\mathrm{M} + O_{\eta \to 0}(\eta^{3/2})\,, \tag{13}$$

*and*

$$\int_{\mathbb{R}^d} \theta \pi_{\mathrm{SGLD}}(\mathrm{d}\theta) - \theta^\star = -(1/2)\nabla^2 U(\theta^\star)^{-1} \mathrm{D}^3 U(\theta^\star)[\mathrm{G}^{-1} \{2\,\mathrm{Id} + (\eta/p)\,\mathrm{M}\}] + O_{\eta \to 0}(\eta^{3/2})\,,$$

$$\int_{\mathbb{R}^d} \theta \pi_{\mathrm{SGD}}(\mathrm{d}\theta) - \theta^\star = -(\eta/2p)\nabla^2 U(\theta^\star)^{-1} \mathrm{D}^3 U(\theta^\star)[\mathrm{G}^{-1}\,\mathrm{M}] + O_{\eta \to 0}(\eta^{3/2})\,,$$

*where*

$$\mathrm{M} = \sum_{i=1}^{N} \left( \nabla U_i(\theta^\star) - \frac{1}{N} \sum_{j=1}^{N} \nabla U_j(\theta^\star) \right)^{\otimes 2}\,, \tag{14}$$

*and* G *is defined in* (9).

*Proof.* The proof is postponed to Section 2.2.2 in the supplementary document. $\square$

Note that this result implies that the mean and the covariance matrix of $\pi_{\mathrm{SGLD}}$ and $\pi_{\mathrm{SGD}}$ stay lower bounded by a positive constant for any $\eta > 0$ as $N \to +\infty$. In Section 4 of the supplementary document, a figure illustrates the results of Theorem 6 and Theorem 7 in the asymptotic $N \to +\infty$.

## 4  Numerical experiments

**Simulated data**  For illustrative purposes, we consider a Bayesian logistic regression in dimension $d = 2$. We simulate $N = 10^5$ covariates $\{x_i\}_{i=1}^{N}$ drawn from a standard 2-dimensional Gaussian distribution and we denote by $\mathbf{X} \in \mathbb{R}^{N \times d}$ the matrix of covariates such that the $i^{\mathrm{th}}$ row of $\mathbf{X}$ is $x_i$. Our Bayesian regression model is specified by a Gaussian prior of mean $0$ and covariance matrix the identity, and a likelihood given for $y_i \in \{0, 1\}$ by $p(y_i|x_i, \theta) = (1 + \mathrm{e}^{-x_i^{\mathrm{T}}\theta})^{-y_i}(1 + \mathrm{e}^{x_i^{\mathrm{T}}\theta})^{y_i - 1}$. We simulate $N$ observations $\{y_i\}_{i=1}^{N}$ under this model. In this setting, **H1** and **H3** are satisfied, and **H2** holds if the state space is compact.

To illustrate the results of Section 3.2, we consider 10 regularly spaced values of $N$ between $10^2$ and $10^5$ and we truncate the dataset accordingly. We compute an estimator $\hat{\theta}$ of $\theta^\star$ using SGD [28]

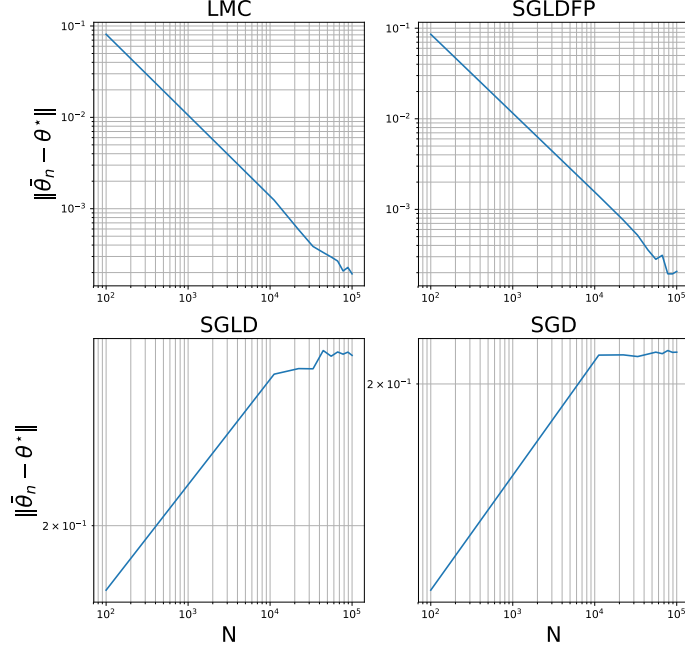

Figure 1: Distance to $\theta^\star$, $\left\|\bar{\theta}_n - \theta^\star\right\|$ for LMC, SGLDFP, SGLD and SGD, function of $N$, in logarithmic scale.

combined with the BFGS algorithm [19]. For the LMC, SGLDFP, SGLD and SGD algorithms, the step size $\gamma$ is set equal to $(1 + \delta/4)^{-1}$ where $\delta$ is the largest eigenvalue of $\mathbf{X}^\mathrm{T}\mathbf{X}$. We start the algorithms at $\theta_0 = \hat{\theta}$ and run $n = 1/\gamma$ iterations where the first $10\%$ samples are discarded as a burn-in period.

We estimate the means and covariance matrices of $\pi_{\mathrm{LMC}}, \pi_{\mathrm{FP}}, \pi_{\mathrm{SGLD}}$ and $\pi_{\mathrm{SGD}}$ by their empirical averages $\bar{\theta}_n = (1/n)\sum_{k=0}^{n-1}\theta_k$ and $\{1/(n-1)\}\sum_{k=0}^{n-1}(\theta_k - \bar{\theta}_n)^{\otimes 2}$. We plot the mean and the trace of the covariance matrices for the different algorithms, averaged over $100$ independent trajectories, in Figure 1 and Figure 2 in logarithmic scale.

The slope for LMC and SGLDFP is $-1$ which confirms the convergence of $\left\|\bar{\theta}_n - \theta^\star\right\|$ to $0$ at a rate $N^{-1}$. On the other hand, we can observe that $\left\|\bar{\theta}_n - \theta^\star\right\|$ converges to a constant for SGD and SGLD.

**Covertype dataset**    We then illustrate our results on the covertype dataset[1] with a Bayesian logistic regression model. The prior is a standard multivariate Gaussian distribution. Given the size of the dataset and the dimension of the problem, LMC requires high computational resources and is not included in the simulations. We truncate the training dataset at $N \in \{10^3, 10^4, 10^5\}$. For all algorithms, the step size $\gamma$ is set equal to $1/N$ and the trajectories are started at $\hat{\theta}$, an estimator of $\theta^\star$, computed using SGD combined with the BFGS algorithm.

We empirically check that the variance of the stochastic gradients scale as $N^2$ for SGD and SGLD, and as $N$ for SGLDFP. We compute the empirical variance estimator of the gradients, take the mean over the dimension and display the result in a logarithmic plot in Figure 3. The slopes are $2$ for SGD and SGLD, and $1$ for SGLDFP.

On the test dataset, we also evaluate the negative loglikelihood of the three algorithms for different values of $N \in \{10^3, 10^4, 10^5\}$, as a function of the number of iterations. The plots are shown in Figure 4. We note that for large $N$, SGLD and SGD give very similar results that are below the performance of SGLDFP.

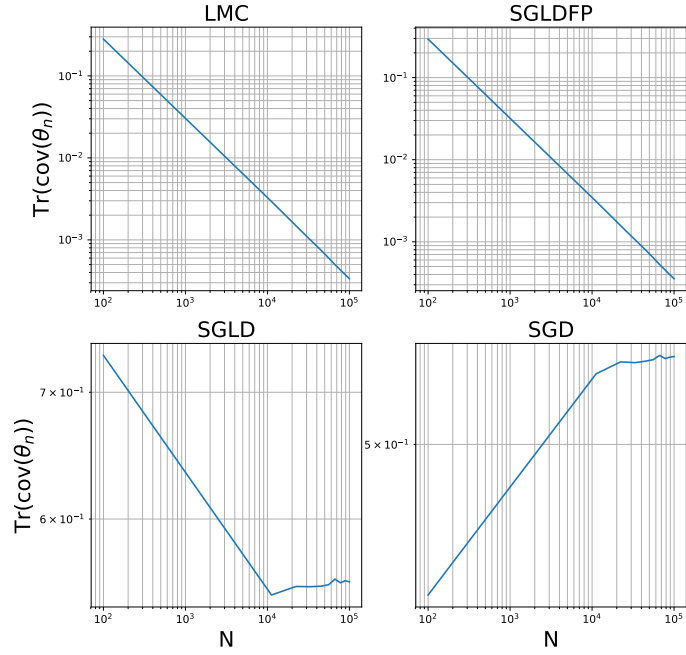

Figure 2: Trace of the covariance matrices for LMC, SGLDFP, SGLD and SGD, function of $N$, in logarithmic scale.

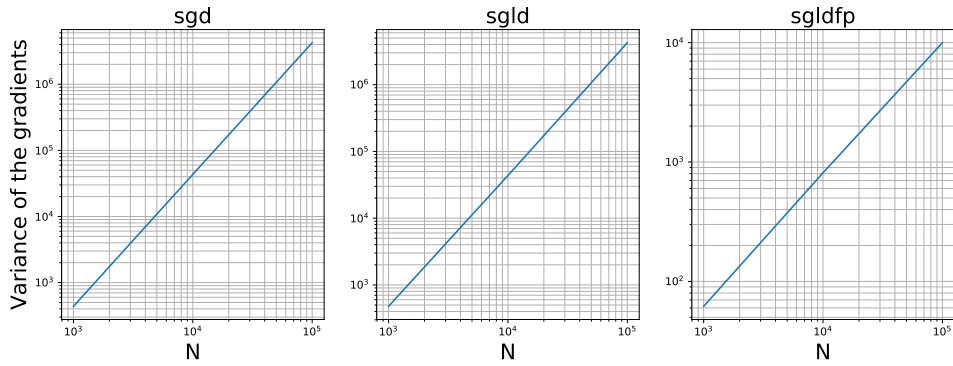

Figure 3: Variance of the stochastic gradients of SGLD, SGLDFP and SGD function of $N$, in logarithmic scale.

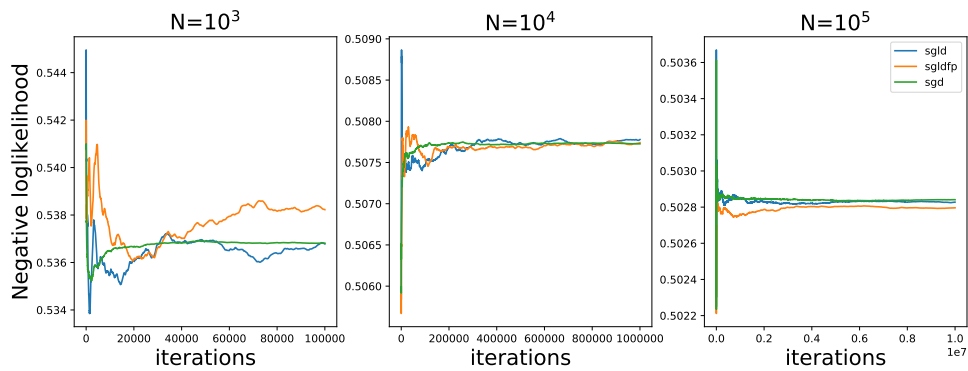

Figure 4: Negative loglikelihood on the test dataset for SGLD, SGLDFP and SGD function of the number of iterations for different values of $N \in \left\{10^3, 10^4, 10^5\right\}$.

## Footnotes

[1]`https://www.csie.ntu.edu.tw/~cjlin/libsvmtools/datasets/binary/covtype.libsvm.binary.scale.bz2`

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
