[Supplementary Material]

# The promises and pitfalls of Stochastic Gradient Langevin Dynamics

## SUPPLEMENTARY DOCUMENT

**Nicolas Brosse, Éric Moulines**
Centre de Mathématiques Appliquées, UMR 7641,
Ecole Polytechnique, Palaiseau, France.
`nicolas.brosse@polytechnique.edu`, `eric.moulines@polytechnique.edu`

**Alain Durmus**
Ecole Normale Supérieure CMLA,
61 Av. du Président Wilson 94235 Cachan Cedex, France.
`alain.durmus@cmla.ens-cachan.fr`

## 1 Proofs of Section 3.1

### 1.1 Proof of Lemma 1

The convergence in Wasserstein distance is classically done via a standard synchronous coupling [Dieuleveut et al., 2017, Proposition 2]. We prove the statement for SGLD; the adaptation for LMC, SGLDFP and SGD is immediate. Let $\gamma \in (0, 2/L)$ and $\lambda_1, \lambda_2 \in \mathcal{P}_2(\mathbb{R}^d)$. By [Villani, 2009, Theorem 4.1], there exists a couple of random variables $(\theta_0^{(1)}, \theta_0^{(2)})$ such that $W_2^2(\lambda_1, \lambda_2) = \mathbb{E}\left[\left\|\theta_0^{(1)} - \theta_0^{(2)}\right\|^2\right]$. Let $(\theta_k^{(1)}, \theta_k^{(2)})_{k \in \mathbb{N}}$ be the SGLD iterates starting from $\theta_0^{(1)}$ and $\theta_0^{(2)}$ respectively and driven by the same noise, *i.e.* for all $k \in \mathbb{N}$,

$$
\begin{cases}
\theta_{k+1}^{(1)} & = \theta_k^{(1)} - \gamma \left\{ \nabla U_0(\theta_k^{(1)}) + (N/p) \sum_{i \in S_{k+1}} \nabla U_i(\theta_k^{(1)}) \right\} + \sqrt{2\gamma} Z_{k+1}\,, \\
\theta_{k+1}^{(2)} & = \theta_k^{(2)} - \gamma \left\{ \nabla U_0(\theta_k^{(2)}) + (N/p) \sum_{i \in S_{k+1}} \nabla U_i(\theta_k^{(2)}) \right\} + \sqrt{2\gamma} Z_{k+1}\,,
\end{cases}
$$

where $(Z_k)_{k \geq 1}$ is an i.i.d. sequence of standard Gaussian variables and $(S_k)_{k \geq 1}$ an i.i.d. sequence of subsamples of $\{1, \ldots, N\}$ of size $p$. Denote by $(\mathcal{F}_k)_{k \in \mathbb{N}}$ the filtration associated to $(\theta_k^{(1)}, \theta_k^{(2)})_{k \in \mathbb{N}}$. We have for $k \in \mathbb{N}$,

$$
\left\|\theta_{k+1}^{(1)} - \theta_{k+1}^{(2)}\right\|^2 =
$$

$$
\left\|\theta_k^{(1)} - \theta_k^{(2)}\right\|^2 + \gamma^2 \left\|\nabla U_0(\theta_k^{(1)}) + \frac{N}{p} \sum_{i \in S_{k+1}} \nabla U_i(\theta_k^{(1)}) - \nabla U_0(\theta_k^{(2)}) - \frac{N}{p} \sum_{i \in S_{k+1}} \nabla U_i(\theta_k^{(2)})\right\|^2
$$

$$
- 2\gamma \left\langle \theta_k^{(1)} - \theta_k^{(2)}, \nabla U_0(\theta_k^{(1)}) + \frac{N}{p} \sum_{i \in S_{k+1}} \nabla U_i(\theta_k^{(1)}) - \nabla U_0(\theta_k^{(2)}) - \frac{N}{p} \sum_{i \in S_{k+1}} \nabla U_i(\theta_k^{(2)})\right\rangle\,.
$$

By **H**1 and **H**3, $\theta \mapsto \nabla U_0(\theta) + (N/p) \sum_{i \in S} \nabla U_i(\theta)$ is $\mathbb{P}$-a.s. $L$-co-coercive Zhu and Marcotte [1996]. Taking the conditional expectation w.r.t. $\mathcal{F}_k$, we obtain

$$
\mathbb{E}\left[\left\|\theta_{k+1}^{(1)} - \theta_{k+1}^{(2)}\right\|^2 \middle| \mathcal{F}_k\right] \leq \left\|\theta_k^{(1)} - \theta_k^{(2)}\right\|^2 - 2\gamma \left\{1 - (\gamma L)/2\right\} \left\langle \theta_k^{(1)} - \theta_k^{(2)}, \nabla U(\theta_k^{(1)}) - \nabla U(\theta_k^{(2)})\right\rangle\,,
$$

and by **H2**

$$\mathbb{E}\left[\left\|\theta_{k+1}^{(1)}-\theta_{k+1}^{(2)}\right\|^{2}\middle|\mathcal{F}_{k}\right] \leq \{1-2m\gamma(1-(\gamma L)/2)\}\left\|\theta_{k}^{(1)}-\theta_{k}^{(2)}\right\|^{2}.$$

Since for all $k \geq 0$, $(\theta_k^{(1)}, \theta_k^{(2)})$ belongs to $\Pi(\lambda_1 R_{\mathrm{SGLD}}^k, \lambda_2 R_{\mathrm{SGLD}}^k)$, we get by a straightforward induction

$$\mathrm{W}_2^2(\lambda_1 R_{\mathrm{SGLD}}^k, \lambda_2 R_{\mathrm{SGLD}}^k) \leq \mathbb{E}\left[\left\|\theta_k^{(1)}-\theta_k^{(2)}\right\|^2\right] \leq \{1-2m\gamma(1-(\gamma L)/2)\}^k \, \mathrm{W}_2^2(\lambda_1, \lambda_2) \,. \quad \text{(S1)}$$

By **H1**, $\lambda_1 R_{\mathrm{SGLD}} \in \mathcal{P}_2(\mathbb{R}^d)$ and taking $\lambda_2 = \lambda_1 R_{\mathrm{SGLD}}$, we get $\sum_{k=0}^{+\infty} \mathrm{W}_2^2(\lambda_1 R_{\mathrm{SGLD}}^k, \lambda_1 R_{\mathrm{SGLD}}^{k+1}) < +\infty$. By [Villani, 2009, Theorem 6.16], $\mathcal{P}_2(\mathbb{R}^d)$ endowed with $\mathrm{W}_2$ is a Polish space. $(\lambda_1 R_{\mathrm{SGLD}}^k)_{k\geq 0}$ is a Cauchy sequence and converges to a limit $\pi_{\mathrm{SGLD}}^{\lambda_1} \in \mathcal{P}_2(\mathbb{R}^d)$. The limit $\pi_{\mathrm{SGLD}}^{\lambda_1}$ does not depend on $\lambda_1$ because, given $\lambda_2 \in \mathcal{P}_2(\mathbb{R}^d)$, by the triangle inequality

$$\mathrm{W}_2(\pi_{\mathrm{SGLD}}^{\lambda_1}, \pi_{\mathrm{SGLD}}^{\lambda_2}) \leq \mathrm{W}_2(\pi_{\mathrm{SGLD}}^{\lambda_1}, \lambda_1 R_{\mathrm{SGLD}}^k) + \mathrm{W}_2(\lambda_1 R_{\mathrm{SGLD}}^k, \lambda_2 R_{\mathrm{SGLD}}^k) + \mathrm{W}_2(\pi_{\mathrm{SGLD}}^{\lambda_2}, \lambda_2 R_{\mathrm{SGLD}}^k) \,.$$

Taking the limit $k \to +\infty$, we get $\mathrm{W}_2(\pi_{\mathrm{SGLD}}^{\lambda_1}, \pi_{\mathrm{SGLD}}^{\lambda_2}) = 0$. The limit is thus the same for all initial distributions and is denoted $\pi_{\mathrm{SGLD}}$. $\pi_{\mathrm{SGLD}}$ is invariant for $R_{\mathrm{SGLD}}$ since we have for all $k \in \mathbb{N}^*$,

$$\mathrm{W}_2(\pi_{\mathrm{SGLD}}, \pi_{\mathrm{SGLD}} R_{\mathrm{SGLD}}) \leq \mathrm{W}_2(\pi_{\mathrm{SGLD}}, \pi_{\mathrm{SGLD}} R_{\mathrm{SGLD}}^k) + \mathrm{W}_2(\pi_{\mathrm{SGLD}} R_{\mathrm{SGLD}}, \pi_{\mathrm{SGLD}} R_{\mathrm{SGLD}}^k) \,.$$

Taking the limit $k \to +\infty$, we obtain $\mathrm{W}_2(\pi_{\mathrm{SGLD}}, \pi_{\mathrm{SGLD}} R_{\mathrm{SGLD}}) = 0$. Using (S1), $\pi_{\mathrm{SGLD}}$ is the unique invariant probability measure for $R_{\mathrm{SGLD}}$.

## 1.2 Proof of Theorem 2

Proof of i). Let $\gamma \in (0, 1/L)$ and $\lambda_1, \lambda_2 \in \mathcal{P}_2(\mathbb{R}^d)$. By [Villani, 2009, Theorem 4.1], there exists a couple of random variables $(\theta_0, \vartheta_0)$ such that $\mathrm{W}_2^2(\lambda_1, \lambda_2) = \mathbb{E}\left[\|\theta_0 - \vartheta_0\|^2\right]$. Let $(\theta_k, \vartheta_k)_{k\in\mathbb{N}}$ be the LMC and SGLDFP iterates starting from $\theta_0$ and $\vartheta_0$ respectively and driven by the same noise, *i.e.* for all $k \in \mathbb{N}$,

$$\begin{cases} \theta_{k+1} &= \theta_k - \gamma \nabla U(\theta_k) + \sqrt{2\gamma} Z_{k+1} \,, \\ \vartheta_{k+1} &= \vartheta_k - \gamma\left(\nabla U_0(\vartheta_k) - \nabla U_0(\theta^\star) + (N/p)\sum_{i\in S_{k+1}}\{\nabla U_i(\vartheta_k) - \nabla U_i(\theta^\star)\}\right) + \sqrt{2\gamma} Z_{k+1} \,, \end{cases}$$

where $(Z_k)_{k\geq 1}$ is an i.i.d. sequence of standard Gaussian variables and $(S_k)_{k\geq 1}$ an i.i.d. sequence of subsamples with replacement of $\{1, \ldots, N\}$ of size $p$. Denote by $(\mathcal{F}_k)_{k\in\mathbb{N}}$ the filtration associated to $(\theta_k, \vartheta_k)_{k\in\mathbb{N}}$. We have for $k \in \mathbb{N}$,

$$\mathbb{E}\left[\|\theta_{k+1} - \vartheta_{k+1}\|^2\middle|\mathcal{F}_k\right] = \|\theta_k - \vartheta_k\|^2 - 2\gamma\langle\theta_k - \vartheta_k, \nabla U(\theta_k) - \nabla U(\vartheta_k)\rangle + \gamma^2 A \quad \text{(S2)}$$

where

$$A = \mathbb{E}\left[\left\|\nabla U(\theta_k) - \left(\nabla U_0(\vartheta_k) - \nabla U_0(\theta^\star) + (N/p)\sum_{i\in S_{k+1}}\{\nabla U_i(\vartheta_k) - \nabla U_i(\theta^\star)\}\right)\right\|^2\middle|\mathcal{F}_k\right]$$

$$= A_1 + A_2 \,,$$

$$A_1 = \|\nabla U(\theta_k) - \nabla U(\vartheta_k)\|^2 \,,$$

$$A_2 = \mathbb{E}\left[\left\|\nabla U(\vartheta_k) - \left(\nabla U_0(\vartheta_k) - \nabla U_0(\theta^\star) + (N/p)\sum_{i\in S_{k+1}}\{\nabla U_i(\vartheta_k) - \nabla U_i(\theta^\star)\}\right)\right\|^2\middle|\mathcal{F}_k\right] \,.$$

Denote by $W$ the random variable equal to $\nabla U_i(\vartheta_k) - \nabla U_i(\theta^\star) - (1/N)\sum_{j=1}^N\{\nabla U_j(\vartheta_k) - \nabla U_j(\theta^\star)\}$ for $i \in \{1, \ldots, N\}$ with probability $1/N$. By **H1** and using the fact that the subsamples $(S_k)_{k\geq 1}$ are drawn with replacement, we obtain

$$A_2 = (N^2/p)\mathbb{E}\left[\|W\|^2\,|\mathcal{F}_k\right] \leq (L^2/p)\|\vartheta_k - \theta^\star\|^2 \,.$$

Combining it with (S2), and using the $L$-co-coercivity of $\nabla U$ under **H**1 and **H**2, we get

$$\mathbb{E}\left[\|\theta_{k+1} - \vartheta_{k+1}\|^2 \middle| \mathcal{F}_k\right] \leq (1 - m\gamma)\|\theta_k - \vartheta_k\|^2 + \{(L^2\gamma^2)/p\}\|\vartheta_k - \theta^\star\|^2 .$$

Iterating and using Lemma S1-i), we have for $n \in \mathbb{N}$

$$\mathrm{W}_2^2(\lambda_1 R_{\mathrm{LMC}}^n, \lambda_2 R_{\mathrm{FP}}^n) \leq \mathbb{E}\left[\|\theta_n - \vartheta_n\|^2\right]$$

$$\leq (1 - m\gamma)^n \, \mathrm{W}_2^2(\lambda_1, \lambda_2) + \frac{L^2\gamma^2}{p}\sum_{k=0}^{n-1}(1 - m\gamma)^{n-1-k}\mathbb{E}\left[\|\vartheta_k - \theta^\star\|^2\right]$$

$$\leq (1 - m\gamma)^n \, \mathrm{W}_2^2(\lambda_1, \lambda_2) + \frac{L^2\gamma^2}{p}n(1 - m\gamma)^{n-1}\int_{\mathbb{R}^d}\|\vartheta - \theta^\star\|^2 \, \lambda_2(\mathrm{d}\vartheta) + \frac{2L^2\gamma d}{pm^2} .$$

Proof of ii). Denote by $\kappa = (2mL)/(m+L)$. By **H**1, **H**2 and [Durmus and Moulines, 2016, Theorem 5], we have for all $n \in \mathbb{N}$,

$$\mathrm{W}_2^2(\lambda_1 P_{n\gamma}, \lambda_2 R_{\mathrm{LMC}}^n) \leq 2\left(1 - \kappa\gamma/2\right)^n \mathrm{W}_2^2(\lambda_1, \lambda_2) + \frac{2L^2\gamma}{\kappa}(\kappa^{-1} + \gamma)\left(2d + \frac{dL^2\gamma^2}{6}\right)$$

$$+ L^4\gamma^3(\kappa^{-1} + \gamma)\sum_{k=1}^{n}\delta_k\left\{1 - \kappa\gamma/2\right\}^{n-k}$$

where for all $k \in \{1, \ldots, n\}$,

$$\delta_k \leq \mathrm{e}^{-2m(k-1)\gamma}\int_{\mathbb{R}^d}\|\vartheta - \theta^\star\|^2 \, \lambda_1(\mathrm{d}\vartheta) + d/m .$$

We get the result by straightforward simplifications and using $\gamma \leq 1/L$.

Proof of iii). Let $\gamma \in (0, 1/L]$ and $\lambda_1, \lambda_2 \in \mathcal{P}_2(\mathbb{R}^d)$. By [Villani, 2009, Thereom 4.1], there exists a couple of random variables $(\theta_0, \vartheta_0)$ such that $\mathrm{W}_2^2(\lambda_1, \lambda_2) = \mathbb{E}\left[\|\theta_0 - \vartheta_0\|^2\right]$. Let $(\theta_k, \vartheta_k)_{k\in\mathbb{N}}$ be the SGLD and SGD iterates starting from $\theta_0$ and $\vartheta_0$ respectively and driven by the same noise, *i.e.* for all $k \in \mathbb{N}$,

$$\begin{cases} \theta_{k+1} &= \theta_k - \gamma\left(\nabla U_0(\theta_k) + (N/p)\sum_{i\in S_{k+1}}\nabla U_i(\theta_k)\right) + \sqrt{2\gamma}Z_{k+1} , \\ \vartheta_{k+1} &= \vartheta_k - \gamma\left(\nabla U_0(\vartheta_k) + (N/p)\sum_{i\in S_{k+1}}\nabla U_i(\vartheta_k)\right) , \end{cases}$$

where $(Z_k)_{k\geq 1}$ is an i.i.d. sequence of standard Gaussian variables and $(S_k)_{k\geq 1}$ an i.i.d. sequence of subsamples with replacement of $\{1, \ldots, N\}$ of size $p$. Denote by $(\mathcal{F}_k)_{k\in\mathbb{N}}$ the filtration associated to $(\theta_k, \vartheta_k)_{k\in\mathbb{N}}$. We have for $k \in \mathbb{N}$,

$$\mathbb{E}\left[\|\theta_{k+1} - \vartheta_{k+1}\|^2 \middle| \mathcal{F}_k\right] = \|\theta_k - \vartheta_k\|^2 - 2\gamma\langle\theta_k - \vartheta_k, \nabla U(\theta_k) - \nabla U(\vartheta_k)\rangle + 2\gamma d$$

$$+ \gamma^2\mathbb{E}\left[\left\|\nabla U_0(\theta_k) + (N/p)\sum_{i\in S_{k+1}}\nabla U_i(\theta_k) - \nabla U_0(\vartheta_k) - (N/p)\sum_{i\in S_{k+1}}\nabla U_i(\vartheta_k)\right\|^2 \middle| \mathcal{F}_k\right] .$$

By **H**1 and **H**3, $\theta \mapsto \nabla U_0(\theta) + (N/p)\sum_{i\in S}\nabla U_i(\theta)$ is $\mathbb{P}$-a.s. $L$-co-coercive and we obtain

$$\mathbb{E}\left[\|\theta_{k+1} - \vartheta_{k+1}\|^2 \middle| \mathcal{F}_k\right] \leq \{1 - 2m\gamma(1 - \gamma L/2)\}\|\theta_k - \vartheta_k\|^2 + 2\gamma d ,$$

which concludes the proof by a straightforward induction.

### 1.3 Proof of Theorem 4

Proof of i). Let $\gamma \in \left(0, \Sigma^{-1}\{1 + N/(p\sum_{i=1}^N x_i^2)\}^{-1}\right]$, $(\theta_k)_{k\in\mathbb{N}}$ be the iterates of SGLD (3) started at $\theta^\star$ and $(\mathcal{F}_k)_{k\in\mathbb{N}}$ the associated filtration. For all $k \in \mathbb{N}$, $\mathbb{E}[\theta_k] = \theta^\star$. The variance of $\theta_k$ satisfies

the following recursion for $k \in \mathbb{N}$

$$\mathbb{E}\left[(\theta_{k+1} - \theta^\star)^2 \big| \mathcal{F}_k\right]$$

$$= \mathbb{E}\left[\left\{\theta_k - \theta^\star - \gamma\left(\Sigma(\theta_k - \theta^\star) + \rho(S_{k+1})(\theta_k - \theta^\star) + \xi(S_{k+1})\right) + \sqrt{2\gamma}Z_{k+1}\right\}^2 \bigg| \mathcal{F}_k\right]$$

$$= \mu(\theta_k - \theta^\star)^2 + 2\gamma + \gamma^2 A \,,$$

where

$$\mu = \mathbb{E}\left[\left\{1 - \gamma\left(\frac{1}{\sigma_\theta^2} + \frac{N}{\sigma_y^2 p}\sum_{i \in S} x_i^2\right)\right\}^2\right] \,, \quad A = \mathbb{E}\left[\left\{\frac{\theta^\star}{\sigma_\theta^2} + \frac{N}{\sigma_y^2 p}\sum_{i \in S}(x_i\theta^\star - y_i)\,x_i\right\}^2\right] \,.$$

We have for $\mu$,

$$\mu = 1 - 2\gamma\Sigma + \gamma^2 \mathbb{E}\left[\left\{\frac{N}{\sigma_y^2 p}\sum_{i \in S}x_i^2 - \frac{1}{\sigma_y^2}\sum_{i=1}^N x_i^2\right\}^2\right] + \gamma^2\Sigma^2$$

$$= 1 - 2\gamma\Sigma + \gamma^2\left\{\Sigma^2 + \frac{N}{\sigma_y^4 p}\sum_{i=1}^N\left(x_i^2 - \frac{1}{N}\sum_{j=1}^N x_j^2\right)\right\} \le 1 - \gamma\Sigma \,,$$

and for $A$,

$$A = \frac{N}{p}\sum_{i=1}^N\left\{\frac{(x_i\theta^\star - y_i)\,x_i}{\sigma_y^2} + \frac{\theta^\star}{N\sigma_\theta^2}\right\}^2 \,.$$

By a straightforward induction, we obtain that the variance of the $n^{\text{th}}$ iterate of SGLD started at $\theta^\star$ is for $n \in \mathbb{N}^*$

$$\int_{\mathbb{R}}(\theta - \theta^\star)^2 R_{\text{SGLD}}^n(\theta^\star, \mathrm{d}\theta) = \frac{1-\mu^n}{1-\mu}2\gamma + \frac{1-\mu^n}{1-\mu}\frac{N\gamma^2}{p}\sum_{i=1}^N\left\{\frac{(x_i\theta^\star - y_i)\,x_i}{\sigma_y^2} + \frac{\theta^\star}{N\sigma_\theta^2}\right\}^2 \,.$$

For SGLDFP, the additive part of the noise in the stochastic gradient disappears and we obtain similarly for $n \in \mathbb{N}^*$

$$\int_{\mathbb{R}}(\theta - \theta^\star)^2 R_{\text{FP}}^n(\theta^\star, \mathrm{d}\theta) = \frac{1-\mu^n}{1-\mu}2\gamma \,.$$

To conclude, we use that for two probability measures with given mean and covariance matrices, the Wasserstein distance between the two Gaussians with these respective parameters is a lower bound for the Wasserstein distance between the two measures [Gelbrich, Theorem 2.1].

The proof of ii) is straightforward.

# 2 Proofs of Section 3.2

## 2.1 Proof of Proposition 5

Let $\theta$ be distributed according to $\pi$. By **H2**, for all $\vartheta \in \mathbb{R}^d$, $U(\vartheta) \ge U(\theta^\star) + (m/2)\|\vartheta - \theta^\star\|^2$ and $\mathbb{E}\left[\nabla U(\theta)\right] = 0$. By a Taylor expansion of $\nabla U$ around $\theta^\star$, we obtain

$$0 = \mathbb{E}\left[\nabla U(\theta)\right] = \nabla^2 U(\theta^\star)\left(\mathbb{E}\left[\theta\right] - \theta^\star\right) + (1/2)\,\mathrm{D}^3\,U(\theta^\star)[\mathbb{E}\left[(\theta - \theta^\star)^{\otimes 2}\right]] + \mathbb{E}\left[\mathcal{R}_1(\theta)\right] \,,$$

where by **H1**, $\mathcal{R}_1 : \mathbb{R}^d \to \mathbb{R}^d$ satisfies

$$\sup_{\vartheta \in \mathbb{R}^d}\left\{\|\mathcal{R}_1(\vartheta)\| / \|\vartheta - \theta^\star\|^3\right\} \le L/6 \,. \tag{S3}$$

Rearranging the terms, we get

$$\mathbb{E}\left[\theta\right] - \theta^\star = -(1/2)\nabla^2 U(\theta^\star)^{-1}\,\mathrm{D}^3\,U(\theta^\star)[\mathbb{E}\left[(\theta - \theta^\star)^{\otimes 2}\right]] - \nabla^2 U(\theta^\star)^{-1}\mathbb{E}\left[\mathcal{R}_1(\theta)\right] \,.$$

To estimate the covariance matrix of $\pi$ around $\theta^\star$, we start again from the Taylor expansion of $\nabla U$ around $\theta^\star$ and we obtain

$$\mathbb{E}\left[\nabla U(\theta)^{\otimes 2}\right] = \mathbb{E}\left[\left(\nabla^2 U(\theta^\star)(\theta - \theta^\star) + \mathcal{R}_2(\theta)\right)^{\otimes 2}\right] = \nabla^2 U(\theta^\star)^{\otimes 2}\mathbb{E}\left[(\theta - \theta^\star)^{\otimes 2}\right] + \mathbb{E}\left[\mathcal{R}_3(\theta)\right] \tag{S4}$$

where by **H**1, $\mathcal{R}_2 : \mathbb{R}^d \to \mathbb{R}^d$ satisfies

$$\sup_{\vartheta \in \mathbb{R}^d}\left\{\|\mathcal{R}_2(\vartheta)\| / \|\vartheta - \theta^\star\|^2\right\} \leq L/2 , \tag{S5}$$

and $\mathcal{R}_3 : \mathbb{R}^d \to \mathbb{R}^{d \times d}$ is defined for all $\vartheta \in \mathbb{R}^d$ by

$$\mathcal{R}_3(\vartheta) = \nabla^2 U(\theta^\star)(\vartheta - \theta^\star) \otimes \mathcal{R}_2(\vartheta) + \mathcal{R}_2(\vartheta) \otimes \nabla^2 U(\theta^\star)(\vartheta - \theta^\star) + \mathcal{R}_2(\vartheta)^{\otimes 2} . \tag{S6}$$

$\mathbb{E}\left[\nabla U(\theta)^{\otimes 2}\right]$ is the Fisher information matrix and by a Taylor expansion of $\nabla^2 U$ around $\theta^\star$ and an integration by parts,

$$\mathbb{E}\left[\nabla U(\theta)^{\otimes 2}\right] = \mathbb{E}\left[\nabla^2 U(\theta)\right] = \nabla^2 U(\theta^\star) + \mathbb{E}\left[\mathcal{R}_4(\theta)\right]$$

where by **H**1, $\mathcal{R}_4 : \mathbb{R}^d \to \mathbb{R}^{d \times d}$ satisfies

$$\sup_{\vartheta \in \mathbb{R}^d}\left\{\|\mathcal{R}_4(\vartheta)\| / \|\vartheta - \theta^\star\|\right\} \leq L . \tag{S7}$$

Combining this result, (S3), (S4), (S5), (S6), (S7) and $\mathbb{E}[\|\theta - \theta^\star\|^4] \leq d(d+2)/m^2$ by [Brosse et al., 2017, Lemma 9] conclude the proof.

## 2.2 Proofs of Theorem 6 and Theorem 7

First note that under **H**1, **H**2 and **H**3, there exists $r \in [0, L/(\sqrt{p}m)]$ such that

$$\mathrm{K} \preceq r^2(\nabla^2 U(\theta^\star))^{\otimes 2} , \tag{S8}$$

*i.e.* for all $A \in \mathbb{R}^{d \times d}$,

$$\mathrm{Tr}(A^{\mathrm{T}}\mathrm{K}(A)) \leq r^2 \mathrm{Tr}(A^{\mathrm{T}}(\nabla^2 U(\theta^\star))^{\otimes 2}A) ,$$

and where K is defined in (7). In addition, if $\liminf_{N \to +\infty} N^{-1}m > 0$, $r$ can be chosen independently of $N$.

Moreover, for all $\gamma \in (0, 2/L)$, H defined in (8), is invertible and for all $\gamma \in (0, 2/\{(1+r^2)L\})$, G defined in (9), is invertible. Indeed,

$$\mathrm{H} = \nabla^2 U(\theta^\star) \otimes \left(\mathrm{Id} - \frac{\gamma}{2}\nabla^2 U(\theta^\star)\right) + \left(\mathrm{Id} - \frac{\gamma}{2}\nabla^2 U(\theta^\star)\right) \otimes \nabla^2 U(\theta^\star) \succ 0 ,$$

$$\mathrm{G} \succeq \nabla^2 U(\theta^\star) \otimes \mathrm{Id} + \mathrm{Id} \otimes \nabla^2 U(\theta^\star) - \gamma(1+r^2)\nabla^2 U(\theta^\star) \otimes \nabla^2 U(\theta^\star)$$

$$\succeq \nabla^2 U(\theta^\star) \otimes \left(\mathrm{Id} - \frac{\gamma(1+r^2)}{2}\nabla^2 U(\theta^\star)\right) + \left(\mathrm{Id} - \frac{\gamma(1+r^2)}{2}\nabla^2 U(\theta^\star)\right) \otimes \nabla^2 U(\theta^\star) \succ 0 .$$

For simplicity of notation, in this Section, we use $\epsilon(\theta)$ to denote the difference between the stochastic and the exact gradients at $\theta \in \mathbb{R}^d$. More precisely, $\epsilon$ is the null function for LMC and is defined for $\theta \in \mathbb{R}^d$ by

$$\epsilon(\theta) = \frac{N}{p}\sum_{i \in S}\nabla U_i(\theta) - \sum_{j=1}^{N}\nabla U_j(\theta) \quad \text{for SGLD and SGD,} \tag{S9}$$

$$\epsilon(\theta) = \nabla U_0(\theta) - \nabla U_0(\theta^\star) + \frac{N}{p}\sum_{i \in S}\left\{\nabla U_i(\theta) - \nabla U_i(\theta^\star)\right\} - \nabla U(\theta) \quad \text{for SGLDFP,} \tag{S10}$$

where $S$ is a random subsample of $\{1, \ldots, N\}$ with replacement of size $p \in \mathbb{N}^*$. In this setting, the update equation for LMC, SGLD and SGLDFP is given for $k \in \mathbb{N}$ by

$$\theta_{k+1} = \theta_k - (\nabla U(\theta_k) + \epsilon_{k+1}(\theta_k)) + \sqrt{2\gamma}Z_{k+1} , \tag{S11}$$

where $(Z_k)_{k \geq 1}$ is a sequence of i.i.d. standard $d$-dimensional Gaussian variables and the sequence of vector fields $(\epsilon_k)_{k \geq 1}$ is associated to a sequence $(S_k)_{k \geq 1}$ of i.i.d. random subsample of $\{1, \ldots, N\}$ with replacement of size $p \in \mathbb{N}^*$. We also denote by $\bar{\pi} \in \mathcal{P}_2(\mathbb{R}^d)$ the invariant probability measure of LMC, SGLDFP or SGLD.

### 2.2.1 Control of the moments of order 2 and 4 of LMC, SGLDFP and SGLD

**Lemma S1.** *Assume **H1**, **H2** and **H3**.*

    *i) For all initial distribution $\lambda \in \mathcal{P}_2(\mathbb{R}^d)$, $\gamma \in (0, 1/L]$ and $k \in \mathbb{N}$,*

$$\mathbb{E}\left[\|\theta_k - \theta^\star\|^2\right] \leq (1 - m\gamma)^k \int_{\mathbb{R}^d} \|\vartheta - \theta^\star\|^2 \, \lambda(\mathrm{d}\vartheta) + (2d)/m$$

    *where $(\theta_k)_{k\in\mathbb{N}}$ are the iterates of SGLDFP (5) or LMC (2).*

    *ii) For all initial distribution $\lambda \in \mathcal{P}_2(\mathbb{R}^d)$, $\gamma \in (0, 1/(2L)]$ and $k \in \mathbb{N}$,*

$$\mathbb{E}\left[\|\theta_k - \theta^\star\|^2\right] \leq (1 - m\gamma)^k \int_{\mathbb{R}^d} \|\vartheta - \theta^\star\|^2 \, \lambda(\mathrm{d}\vartheta) + \frac{2d}{m}$$

$$+ \frac{2\gamma N}{mp} \sum_{i=1}^{N} \left\| \nabla U_i(\theta^\star) - \frac{1}{N}\sum_{j=1}^{N} \nabla U_j(\theta^\star) \right\|^2$$

    *where $(\theta_k)_{k\in\mathbb{N}}$ are the iterates of SGLD (3).*

*Proof.* i). We prove the result for SGLDFP, the case of LMC is identical. Let $\gamma \in (0, 1/L]$, $(\theta_k)_{k\in\mathbb{N}}$ be the iterates of SGLDFP and $(\mathcal{F}_k)_{k\in\mathbb{N}}$ the filtration associated to $(\theta_k)_{k\in\mathbb{N}}$. By (5), we have for all $k \in \mathbb{N}$,

$$\mathbb{E}\left[\|\theta_{k+1} - \theta^\star\|^2 \Big| \mathcal{F}_k\right] = \|\theta_k - \theta^\star\|^2 - 2\gamma \langle \theta_k - \theta^\star, \nabla U(\theta_k) - \nabla U(\theta^\star)\rangle + 2\gamma d$$

$$+ \gamma^2 \mathbb{E}\left[\left\| \nabla U_0(\theta_k) - \nabla U_0(\theta^\star) + \frac{N}{p}\sum_{i\in S_{k+1}} \{\nabla U_i(\theta_k) - \nabla U_i(\theta^\star)\} \right\|^2 \Bigg| \mathcal{F}_k \right]$$

By **H**1 and **H**3, $\theta \mapsto \nabla U_0(\theta) - \nabla U_0(\theta^\star) + (N/p)\sum_{i\in S}\{\nabla U_i(\theta) - \nabla U_i(\theta^\star)\}$ is $\mathbb{P}$-a.s. $L$-co-coercive and we obtain

$$\mathbb{E}\left[\|\theta_{k+1} - \theta^\star\|^2 \Big| \mathcal{F}_k\right] \leq \{1 - 2m\gamma(1 - \gamma L/2)\} \|\theta_k - \theta^\star\|^2 + 2\gamma d \,.$$

A straightforward induction concludes the proof.

ii). Let $\gamma \in (0, 1/(2L)]$, $(\theta_k)_{k\in\mathbb{N}}$ be the iterates of SGLD and $(\mathcal{F}_k)_{k\in\mathbb{N}}$ the filtration associated to $(\theta_k)_{k\in\mathbb{N}}$. By (3), we have for all $k \in \mathbb{N}$,

$$\mathbb{E}\left[\|\theta_{k+1} - \theta^\star\|^2 \Big| \mathcal{F}_k\right] = \|\theta_k - \theta^\star\|^2 - 2\gamma \langle \theta_k - \theta^\star, \nabla U(\theta_k) - \nabla U(\theta^\star)\rangle + 2\gamma d$$

$$+ \gamma^2 \mathbb{E}\left[\left\| \nabla U_0(\theta_k) + \frac{N}{p}\sum_{i\in S_{k+1}} \nabla U_i(\theta_k) \right\|^2 \Bigg| \mathcal{F}_k \right]$$

$$\leq \|\theta_k - \theta^\star\|^2 - 2\gamma \langle \theta_k - \theta^\star, \nabla U(\theta_k) - \nabla U(\theta^\star)\rangle + 2\gamma d$$

$$+ 2\gamma^2 \mathbb{E}\left[\left\| \nabla U_0(\theta_k) - \nabla U_0(\theta^\star) + \frac{N}{p}\sum_{i\in S_{k+1}} \{\nabla U_i(\theta_k) - \nabla U_i(\theta^\star)\} \right\|^2 \Bigg| \mathcal{F}_k \right]$$

$$+ 2\gamma^2 \mathbb{E}\left[\left\| \nabla U_0(\theta^\star) + \frac{N}{p}\sum_{i\in S_{k+1}} \nabla U_i(\theta^\star) \right\|^2 \Bigg| \mathcal{F}_k \right] \,.$$

By **H**1 and **H**3, $\theta \mapsto \nabla U_0(\theta) + (N/p)\sum_{i\in S} \nabla U_i(\theta)$ is $\mathbb{P}$-a.s. $L$-co-coercive and we obtain

$$\mathbb{E}\left[\|\theta_{k+1} - \theta^\star\|^2 \Big| \mathcal{F}_k\right] \leq \{1 - 2m\gamma(1 - \gamma L)\} \|\theta_k - \theta^\star\|^2 + 2\gamma d$$

$$+ \frac{2\gamma^2 N}{p} \sum_{i=1}^{N} \left\| \nabla U_i(\theta^\star) - \frac{1}{N}\sum_{j=1}^{N} \nabla U_j(\theta^\star) \right\|^2 \,.$$

A straightforward induction concludes the proof. $\qquad\square$

**Lemma S2.** *Assume **H1**, **H2** and **H3**. For all initial distribution $\lambda \in \mathcal{P}_4(\mathbb{R}^d)$, $\gamma \in (0, 1/\{12(L \vee 1)\}]$ and $k \in \mathbb{N}$,*

$$
\mathbb{E}\left[\|\theta_k - \theta^\star\|^4\right] \leq (1 - 2m\gamma)^k \int_{\mathbb{R}^d} \|\vartheta - \theta^\star\|^4 \, \lambda(\mathrm{d}\vartheta)
$$

$$
+ \left\{ 12\gamma^2 \mathbb{E}\left[\|\epsilon(\theta^\star)\|^2\right] + 2\gamma(2d+1) \right\} k(1 - m\gamma)^{k-1} \int_{\mathbb{R}^d} \|\vartheta - \theta^\star\|^2 \, \lambda(\mathrm{d}\vartheta)
$$

$$
+ \left\{ \frac{2d+1}{m} + \frac{6\gamma}{m} \mathbb{E}\left[\|\epsilon(\theta^\star)\|^2\right] \right\}^2
$$

$$
+ \frac{2\gamma d(2+d)}{m} + \frac{4\gamma^3}{m} \mathbb{E}\left[\|\epsilon(\theta^\star)\|^4\right] + \frac{4\gamma^2(d+2)}{m} \mathbb{E}\left[\|\epsilon(\theta^\star)\|^2\right] \ .
$$

*where $(\theta_k)_{k\in\mathbb{N}}$ are the iterates of LMC (2), SGLD (3) or SGLDFP (5).*

*Proof.* Let $\gamma \in (0, 1/\{12(L \vee 1)\}]$, $(\theta_k)_{k\in\mathbb{N}}$ be the iterates of LMC (2), SGLD (3) or SGLDFP (5) and $(\mathcal{F}_k)_{k\in\mathbb{N}}$ be the associated filtration. By developing the square, we have

$$
\|\theta_1 - \theta^\star\|^4 = \Big( \|\theta_0 - \theta^\star\|^2 + 2\gamma \|Z_1\|^2 + \gamma^2 \|\nabla U(\theta_0) + \epsilon_1(\theta_0)\|^2
$$

$$
- 2\gamma \langle \nabla U(\theta_0) + \epsilon_1(\theta_0), \theta_0 - \theta^\star \rangle + \sqrt{2\gamma} \langle \theta_0 - \theta^\star, Z_1 \rangle - (2\gamma)^{3/2} \langle \nabla U(\theta_0) + \epsilon_1(\theta_0), Z_1 \rangle \Big)^2 ,
$$

and taking the conditional expectation w.r.t. $\mathcal{F}_0$,

$$
\mathbb{E}\left[\|\theta_1 - \theta^\star\|^4 \Big| \mathcal{F}_0\right] = \mathbb{E}\Big[ \|\theta_0 - \theta^\star\|^4 + 4\gamma^2 \|Z_1\|^4 + \gamma^4 \|\nabla U(\theta_0) + \epsilon_1(\theta_0)\|^4
$$

$$
+ 4\gamma^2 \langle \nabla U(\theta_0) + \epsilon_1(\theta_0), \theta_0 - \theta^\star \rangle^2 + 2\gamma \langle \theta_0 - \theta^\star, Z_1 \rangle^2 + (2\gamma)^3 \langle \nabla U(\theta_0) + \epsilon_1(\theta_0), Z_1 \rangle^2
$$

$$
+ 4\gamma \|Z_1\|^2 \|\theta_0 - \theta^\star\|^2 + 2\gamma^2 \|\theta_0 - \theta^\star\|^2 \|\nabla U(\theta_0) + \epsilon_1(\theta_0)\|^2
$$

$$
- 4\gamma \|\theta_0 - \theta^\star\|^2 \langle \nabla U(\theta_0), \theta_0 - \theta^\star \rangle + 4\gamma^3 \|Z_1\|^2 \|\nabla U(\theta_0) + \epsilon_1(\theta_0)\|^2
$$

$$
- 8\gamma^2 \|Z_1\|^2 \langle \nabla U(\theta_0), \theta_0 - \theta^\star \rangle - 4\gamma^3 \|\nabla U(\theta_0) + \epsilon_1(\theta_0)\|^2 \langle \nabla U(\theta_0) + \epsilon_1(\theta_0), \theta_0 - \theta^\star \rangle
$$

$$
- 8\gamma^2 \langle \theta_0 - \theta^\star, Z_1 \rangle \langle \nabla U(\theta_0) + \epsilon_1(\theta_0), Z_1 \rangle \, |\mathcal{F}_0\Big] \ .
$$

By **H1** and **H3**, $\theta \mapsto \nabla U(\theta) + \epsilon_1(\theta)$ is $\mathbb{P}$-a.s. $L$-co-coercive and we have for all $\theta \in \mathbb{R}^d$, $\mathbb{P}$-a.s.,

$$
\|\nabla U(\theta) + \epsilon_1(\theta) - \epsilon_1(\theta^\star)\|^2 \leq L \langle \theta - \theta^\star, \nabla U(\theta) + \epsilon_1(\theta) - \epsilon_1(\theta^\star) \rangle \ ,
$$

$$
\|\nabla U(\theta) + \epsilon_1(\theta) - \epsilon_1(\theta^\star)\|^4 \leq L^2 \|\theta - \theta^\star\|^2 \langle \theta - \theta^\star, \nabla U(\theta) + \epsilon_1(\theta) - \epsilon_1(\theta^\star) \rangle \ .
$$

Combining it with $\mathbb{E}\left[\|Z_1\|^4\right] = d(2+d)$, we obtain

$$
\mathbb{E}\left[\|\theta_1 - \theta^\star\|^4 \Big| \mathcal{F}_0, S_1\right] \leq \|\theta_0 - \theta^\star\|^4 - 4\gamma(1 - 3\gamma L - 2\gamma^3 L^2) \|\theta_0 - \theta^\star\|^2
$$

$$
\times \langle \theta_0 - \theta^\star, \nabla U(\theta_0) + \epsilon_1(\theta_0) - \epsilon_1(\theta^\star) \rangle + (12\gamma^2 \|\epsilon_1(\theta^\star)\|^2 + 2\gamma(2d+1)) \|\theta_0 - \theta^\star\|^2
$$

$$
+ 4\gamma^2 d(2+d) + 8\gamma^4 \|\epsilon_1(\theta^\star)\|^4 + 8\gamma^3(d+2) \|\epsilon_1(\theta^\star)\|^2
$$

$$
- 8(d+1)\gamma^2(1 - 2\gamma L) \langle \theta_0 - \theta^\star, \nabla U(\theta_0) + \epsilon_1(\theta_0) - \epsilon_1(\theta^\star) \rangle \ .
$$

By **H2** and using $\gamma \leq 1/\{12(L \vee 1)\}$, we get

$$
\mathbb{E}\left[\|\theta_1 - \theta^\star\|^4 \Big| \mathcal{F}_0\right] \leq (1 - 2m\gamma) \|\theta_0 - \theta^\star\|^4 + \left\{ 12\gamma^2 \mathbb{E}\left[\|\epsilon_1(\theta^\star)\|^2\right] + 2\gamma(2d+1) \right\} \|\theta_0 - \theta^\star\|^2
$$

$$
+ 4\gamma^2 d(2+d) + 8\gamma^4 \mathbb{E}\left[\|\epsilon_1(\theta^\star)\|^4\right] + 8\gamma^3(d+2)\mathbb{E}\left[\|\epsilon_1(\theta^\star)\|^2\right] \ .
$$

By a straightforward induction, we have for all $n \in \mathbb{N}$

$$
\mathbb{E}\left[\|\theta_n - \theta^\star\|^4\right] \leq (1 - 2m\gamma)^n \mathbb{E}\left[\|\theta_0 - \theta^\star\|^4\right]
$$

$$
+ \left\{ 12\gamma^2 \mathbb{E}\left[\|\epsilon(\theta^\star)\|^2\right] + 2\gamma(2d+1) \right\} \sum_{k=0}^{n-1} (1 - 2m\gamma)^{n-1-k} \mathbb{E}\left[\|\theta_k - \theta^\star\|^2\right]
$$

$$
+ (2m\gamma)^{-1} \left\{ 4\gamma^2 d(2+d) + 8\gamma^4 \mathbb{E}\left[\|\epsilon(\theta^\star)\|^4\right] + 8\gamma^3(d+2)\mathbb{E}\left[\|\epsilon(\theta^\star)\|^2\right] \right\}
$$

and by Lemma S1,

$$\mathbb{E}\left[\|\theta_n - \theta^\star\|^4\right] \leq (1 - 2m\gamma)^n \int_{\mathbb{R}^d} \|\vartheta - \theta^\star\|^4 \,\lambda(\mathrm{d}\vartheta)$$

$$+ \left\{12\gamma^2 \mathbb{E}\left[\|\epsilon_1(\theta^\star)\|^2\right] + 2\gamma(2d+1)\right\} n(1 - m\gamma)^{n-1} \int_{\mathbb{R}^d} \|\vartheta - \theta^\star\|^2 \,\lambda(\mathrm{d}\vartheta)$$

$$+ \left\{\frac{2d+1}{m} + \frac{6\gamma}{m}\mathbb{E}\left[\|\epsilon(\theta^\star)\|^2\right]\right\}^2$$

$$+ \frac{2\gamma d(2+d)}{m} + \frac{4\gamma^3}{m}\mathbb{E}\left[\|\epsilon(\theta^\star)\|^4\right] + \frac{4\gamma^2(d+2)}{m}\mathbb{E}\left[\|\epsilon(\theta^\star)\|^2\right] \ .$$

$\square$

Thanks to this lemma, we obtain the following corollary. The upper bound for SGD is given by [Dieuleveut et al., 2017, Lemma 13].

**Corollary 3.** *Assume H1, H2 and H3.*

    *i) Let $\gamma = \eta/N$ with $\eta \in (0, 1/\{24(\tilde{L} \vee 1)\}]$ and assume that $\liminf_{N \to +\infty} N^{-1}m > 0$. Then,*

$$\int_{\mathbb{R}^d} \|\theta - \theta^\star\|^4 \,\pi_{\mathrm{LMC}}(\mathrm{d}\theta) = d^2 O_{N \to +\infty}(N^{-2}) \ ,$$

$$\int_{\mathbb{R}^d} \|\theta - \theta^\star\|^4 \,\pi_{\mathrm{FP}}(\mathrm{d}\theta) = d^2 O_{N \to +\infty}(N^{-2}) \ .$$

    *ii) Let $\gamma = \eta/N$ with $\eta \in (0, 1/\{24(\tilde{L} \vee 1)\}]$ and assume that $\liminf_{N \to +\infty} N^{-1}m > 0$ and that $N \geq 1/\eta$. Then,*

$$\int_{\mathbb{R}^d} \|\theta - \theta^\star\|^4 \,\pi_{\mathrm{SGLD}}(\mathrm{d}\theta) = d^2 O_{\eta \to 0}(\eta^2) \ , \quad \int_{\mathbb{R}^d} \|\theta - \theta^\star\|^4 \,\pi_{\mathrm{SGD}}(\mathrm{d}\theta) = d^2 O_{\eta \to 0}(\eta^2) \ .$$

### 2.2.2 Proofs of Theorem 6 and Theorem 7

Denote by

$$\eta_0 = \inf_{N \geq 1} \left\{\frac{N}{12(L \vee 1)} \wedge \frac{2N}{(1+r^2)L}\right\} > 0 \ , \tag{S12}$$

and set $\gamma = \eta/N$ with $\eta \in (0, \eta_0)$. Let $\delta \in \{0, 1\}$ be equal to 1 for LMC, SGLDFP and SGLD and 0 for SGD. Let $\theta_0$ be distributed according to $\bar{\pi}$. By (S11) and using a Taylor expansion around $\theta^\star$ for $\nabla U$, we obtain

$$\theta_1 - \theta^\star = \theta_0 - \theta^\star - \gamma\left(\nabla^2 U(\theta^\star)(\theta_0 - \theta^\star) + \mathcal{R}_1(\theta_0) + \epsilon_1(\theta_0)\right) + \delta\sqrt{2\gamma}Z_1 \ ,$$

where by **H1**, $\mathcal{R}_1 : \mathbb{R}^d \to \mathbb{R}^d$ satisfies

$$\sup_{\theta \in \mathbb{R}^d} \left\{\|\mathcal{R}_1(\theta)\| \,/\, \|\theta - \theta^\star\|^2\right\} \leq L/2 \ . \tag{S13}$$

Taking the tensor product and the expectation, and using that $\theta_0, \epsilon_1, Z_1$ are mutually independent, we obtain

$$\mathrm{H}\,\mathbb{E}\left[(\theta_0 - \theta^\star)^{\otimes 2}\right] = 2\delta\,\mathrm{Id} + \gamma\mathbb{E}\left[\epsilon_1(\theta_0)^{\otimes 2}\right] + \mathbb{E}\left[\mathcal{R}_1(\theta_0) \otimes \{\theta_0 - \theta^\star\} + \{\theta_0 - \theta^\star\} \otimes \mathcal{R}_1(\theta_0)\right]$$

$$+ \gamma\mathbb{E}\left[\mathcal{R}_1(\theta_0)^{\otimes 2} + \{\nabla^2 U(\theta^\star)(\theta_0 - \theta^\star)\} \otimes \mathcal{R}_1(\theta_0) + \mathcal{R}_1(\theta_0) \otimes \nabla^2 U(\theta^\star)(\theta_0 - \theta^\star)\right] \ . \tag{S14}$$

For LMC, $\epsilon_1$ is the null function and by Corollary 3-i), (S13) and (S14), we obtain (10). Regarding SGLDFP, SGLD and SGD, by a Taylor expansion of $\epsilon_1$ around $\theta^\star$, we get for all $\theta \in \mathbb{R}^d$, $\mathbb{P}$-a.s. ,

$$\epsilon_1(\theta) = \epsilon_1(\theta^\star) + \nabla\epsilon_1(\theta^\star)(\theta - \theta^\star) + \mathcal{R}_2(\theta)$$

where by **H1**, $\mathcal{R}_2 : \mathbb{R}^d \to \mathbb{R}^d$ satisfies

$$\sup_{\theta \in \mathbb{R}^d} \left\{\|\mathcal{R}_2(\theta)\| \,/\, \|\theta - \theta^\star\|^2\right\} \leq L/2 \ . \tag{S15}$$

Therefore, taking the tensor product and the expectation, we obtain

$$\mathbb{E}\left[\epsilon_1(\theta_0)^{\otimes 2}\right] = \mathbb{E}\left[\epsilon_1(\theta^\star)^{\otimes 2}\right] + (\nabla\epsilon_1(\theta^\star))^{\otimes 2}\,\mathbb{E}\left[(\theta_0 - \theta^\star)^{\otimes 2}\right] + \mathbb{E}\left[\mathcal{R}_3(\theta_0)\right] \qquad (S16)$$

where $\mathcal{R}_3 : \mathbb{R}^d \to \mathbb{R}^{d\times d}$ is defined for all $\theta \in \mathbb{R}^d$, $\mathbb{P}$-a.s. ,

$$\begin{aligned}
\mathcal{R}_3(\theta) &= \epsilon_1(\theta^\star) \otimes \{\nabla\epsilon_1(\theta^\star)(\theta - \theta^\star)\} + \{\nabla\epsilon_1(\theta^\star)(\theta - \theta^\star)\} \otimes \epsilon_1(\theta^\star) \\
&\quad + \{\epsilon_1(\theta^\star) + \nabla\epsilon_1(\theta^\star)(\theta - \theta^\star)\} \otimes \mathcal{R}_2(\theta) + \mathcal{R}_2(\theta) \otimes \{\epsilon_1(\theta^\star) + \nabla\epsilon_1(\theta^\star)(\theta - \theta^\star)\} + \mathcal{R}_2^{\otimes 2}(\theta) .
\end{aligned} \qquad (S17)$$

Note that $\mathrm{K} = \mathbb{E}\left[(\nabla\epsilon_1(\theta^\star))^{\otimes 2}\right]$. For SGLDFP, $\epsilon_1(\theta^\star) = 0$ a.s. By Corollary 3-i), (S13), (S14), (S15), (S16) and (S17), we obtain (11).

Regarding SGLD and SGD, we have $\mathbb{E}\left[\epsilon_1(\theta^\star)^{\otimes 2}\right] = (N/p)\,\mathrm{M}$ where M is defined in (14). By Corollary 3-ii), (S13), (S14), (S15), (S16) and (S17), we obtain (12) and (13).

For the mean of $\pi_{\mathrm{LMC}}, \pi_{\mathrm{FP}}, \pi_{\mathrm{SGLD}}$ and $\pi_{\mathrm{SGD}}$, by a Taylor expansion around $\theta^\star$ for $\nabla U$ of order 3, we obtain

$$\begin{aligned}
\theta_1 - \theta^\star = \theta_0 - \theta^\star - \gamma\left(\nabla^2 U(\theta^\star)(\theta_0 - \theta^\star) + (1/2)\,\mathrm{D}^3\,U(\theta^\star)(\theta_0 - \theta^\star)^{\otimes 2} + \mathcal{R}_4(\theta_0) + \epsilon_1(\theta_0)\right) \\
+ \delta\sqrt{2\gamma}Z_1 ,
\end{aligned}$$

where by **H**1, $\mathcal{R}_4 : \mathbb{R}^d \to \mathbb{R}^d$ satisfies

$$\sup_{\theta\in\mathbb{R}^d}\left\{\|\mathcal{R}_4(\theta)\| / \|\theta - \theta^\star\|^3\right\} \le L/6 . \qquad (S18)$$

Taking the expectation and using that $\theta_1$ is distributed according to $\bar{\pi}$, we get

$$\mathbb{E}[\theta_0] - \theta^\star = -(1/2)\nabla^2 U(\theta^\star)\,\mathrm{D}^3\,U(\theta^\star)[\mathbb{E}\left[(\theta_0 - \theta^\star)^{\otimes 2}\right]] - \nabla^2 U(\theta^\star)^{-1}\mathbb{E}\left[\mathcal{R}_4(\theta_0)\right] .$$

(10), (11), (12),(13), (S18) and Corollary 3 conclude the proof.

# 3 Means and covariance matrices of $\pi_{\mathrm{LMC}}, \pi_{\mathrm{FP}}, \pi_{\mathrm{SGLD}}$ and $\pi_{\mathrm{SGD}}$ in the Bayesian linear regression

In this Section, we provide explicit expressions of the covariance matrices of $\pi_{\mathrm{LMC}}, \pi_{\mathrm{FP}}, \pi_{\mathrm{SGLD}}$ and $\pi_{\mathrm{SGD}}$ in the context of the Bayesian linear regression. In this setting, the algorithms are without bias, *i.e.*

$$\int_{\mathbb{R}^d}\theta\pi_{\mathrm{LMC}}(\mathrm{d}\theta) = \int_{\mathbb{R}^d}\theta\pi_{\mathrm{FP}}(\mathrm{d}\theta) = \int_{\mathbb{R}^d}\theta\pi_{\mathrm{SGLD}}(\mathrm{d}\theta) = \int_{\mathbb{R}^d}\theta\pi_{\mathrm{SGD}}(\mathrm{d}\theta) = \int_{\mathbb{R}^d}\theta\pi(\mathrm{d}\theta) = \theta^\star . \qquad (S19)$$

Before giving the expressions of the variances in Theorem S4, we define $\mathrm{T} : \mathbb{R}^{d\times d} \to \mathbb{R}^{d\times d}$ for all $A \in \mathbb{R}^{d\times d}$ by

$$\mathrm{T}(A) = \mathbb{E}\left[\left(\frac{\mathrm{Id}}{\sigma_\theta^2} + \frac{N}{p\sigma_y^2}\sum_{i\in S}x_i x_i^{\mathrm{T}} - \Sigma\right)^{\otimes 2} A\right] = \frac{N}{p}\sum_{i=1}^N\left(\frac{x_i x_i^{\mathrm{T}}}{\sigma_y^2} + \frac{\mathrm{Id}}{N\sigma_\theta^2} - \frac{\Sigma}{N}\right)^{\otimes 2} A , \qquad (S20)$$

where $S$ is a random subsample of $\{1,\dots,N\}$ with replacement of size $p \in \mathbb{N}^\star$. Note that, in this setting, $\tilde{L} = \max_{i\in\{1,\dots,N\}}\|x_i\|^2$ and $m$ is the smallest eigenvalue of $\Sigma$. There exists $r \in [0, L/(\sqrt{p}m)]$ such that

$$\mathrm{T} \preceq r^2\Sigma^{\otimes 2} \qquad (S21)$$

*i.e.* for all $A \in \mathbb{R}^{d\times d}$, $\mathrm{Tr}(A^{\mathrm{T}}\,\mathrm{T}\cdot A) \le r^2\,\mathrm{Tr}(A^{\mathrm{T}}\Sigma^{\otimes 2}A)$. Assuming that $\liminf_{N\to+\infty}N^{-1}m > 0$, $r$ can be chosen independently of $N$.

**Theorem S4.** *Consider the case of the Bayesian linear regression. We have for all $\gamma \in (0, 2/L)$*

$$\int_{\mathbb{R}^d}(\theta - \theta^\star)^{\otimes 2}\pi_{\mathrm{LMC}}(\mathrm{d}\theta) = (\mathrm{Id}\otimes\Sigma + \Sigma\otimes\mathrm{Id} - \gamma\Sigma\otimes\Sigma)^{-1}(2\,\mathrm{Id}) ,$$

*and for all* $\gamma \in \left(0, 2/\{(1+r^2)L\}\right)$,

$$\int_{\mathbb{R}^d} (\theta - \theta^\star)^{\otimes 2} \pi_{\mathrm{FP}}(\mathrm{d}\theta) = \left\{\mathrm{Id}\otimes\Sigma + \Sigma\otimes\mathrm{Id} - \gamma(\Sigma^{\otimes 2} + \mathrm{T})\right\}^{-1}(2\,\mathrm{Id}),$$

$$\int_{\mathbb{R}^d} (\theta - \theta^\star)^{\otimes 2} \pi_{\mathrm{SGLD}}(\mathrm{d}\theta) = \left\{\mathrm{Id}\otimes\Sigma + \Sigma\otimes\mathrm{Id} - \gamma(\Sigma^{\otimes 2} + \mathrm{T})\right\}^{-1}$$
$$\cdot \left\{2\,\mathrm{Id} + \frac{\gamma N}{p}\sum_{i=1}^{N}\left(\frac{(x_i^{\mathrm{T}}\theta^\star - y_i)x_i}{\sigma_y^2} + \frac{\theta^\star}{\sigma_\theta^2}\right)^{\otimes 2}\right\},$$

$$\int_{\mathbb{R}^d} (\theta - \theta^\star)^{\otimes 2} \pi_{\mathrm{SGD}}(\mathrm{d}\theta) = \left\{\mathrm{Id}\otimes\Sigma + \Sigma\otimes\mathrm{Id} - \gamma(\Sigma^{\otimes 2} + \mathrm{T})\right\}^{-1}$$
$$\cdot \frac{\gamma N}{p}\sum_{i=1}^{N}\left(\frac{(x_i^{\mathrm{T}}\theta^\star - y_i)x_i}{\sigma_y^2} + \frac{\theta^\star}{\sigma_\theta^2}\right)^{\otimes 2}.$$

*Proof.* We prove the result for SGLD, the adaptation to the other algorithms is immediate. Let $\gamma \in \left(0, 2/\{(1+r^2)L\}\right)$, $\theta_0$ be distributed according to $\pi_{\mathrm{SGLD}}$ and $\theta_1$ be given by (3). By definition of $\pi_{\mathrm{SGLD}}$, $\theta_1$ is distributed according to $\pi_{\mathrm{SGLD}}$. We have

$$\mathbb{E}\left[(\theta_1 - \theta^\star)^{\otimes 2}\right] = \mathbb{E}\left[\left[\left\{\mathrm{Id} - \gamma\left(\frac{\mathrm{Id}}{\sigma_\theta^2} + \frac{N}{p\sigma_y^2}\sum_{i\in S_1} x_i x_i^{\mathrm{T}}\right)\right\}(\theta_0 - \theta^\star)\right.\right.$$
$$\left.\left. - \gamma\left(\frac{\theta^\star}{\sigma_\theta^2} + \frac{N}{p\sigma_y^2}\sum_{i\in S_1}(x_i^{\mathrm{T}}\theta^\star - y_i)x_i\right) + \sqrt{2\gamma}Z_1\right]^{\otimes 2}\right].$$

Using that $\theta_0, S_1, Z_1$ are mutually independent, we obtain

$$\left\{\mathrm{Id}\otimes\Sigma + \Sigma\otimes\mathrm{Id} - \gamma\mathbb{E}\left[\left(\frac{\mathrm{Id}}{\sigma_\theta^2} + \frac{N}{p\sigma_y^2}\sum_{i\in S_1} x_i x_i^{\mathrm{T}}\right)^{\otimes 2}\right]\right\}\mathbb{E}\left[(\theta_0 - \theta^\star)^{\otimes 2}\right]$$
$$= 2\,\mathrm{Id} + \gamma\mathbb{E}\left[\left(\frac{\theta^\star}{\sigma_\theta^2} + \frac{N}{p\sigma_y^2}\sum_{i\in S_1}(x_i^{\mathrm{T}}\theta^\star - y_i)x_i\right)^{\otimes 2}\right]$$

and

$$\left\{\mathrm{Id}\otimes\Sigma + \Sigma\otimes\mathrm{Id} - \gamma(\Sigma^{\otimes 2} + \mathrm{T})\right\}\mathbb{E}\left[(\theta_0 - \theta^\star)^{\otimes 2}\right]$$
$$= 2\,\mathrm{Id} + \frac{\gamma N}{p}\sum_{i=1}^{N}\left(\frac{(x_i^{\mathrm{T}}\theta^\star - y_i)x_i}{\sigma_y^2} + \frac{\theta^\star}{\sigma_\theta^2}\right)^{\otimes 2}.$$

On $\mathbb{R}^{d\times d}$ equipped with the Hilbert-Schmidt inner product, $\mathrm{Id}\otimes\Sigma + \Sigma\otimes\mathrm{Id} - \gamma(\Sigma^{\otimes 2} + \mathrm{T})$ is a positive definite operator. Indeed, by (S21),

$$\mathrm{Id}\otimes\Sigma + \Sigma\otimes\mathrm{Id} - \gamma(\Sigma^{\otimes 2} + \mathrm{T}) \succeq \mathrm{Id}\otimes\Sigma + \Sigma\otimes\mathrm{Id} - \gamma(1+r^2)\Sigma^{\otimes 2}$$
$$= \left(\mathrm{Id} - \gamma\frac{1+r^2}{2}\Sigma\right)\otimes\Sigma + \Sigma\otimes\left(\mathrm{Id} - \gamma\frac{1+r^2}{2}\Sigma\right) \succ 0$$

for $\gamma \in \left(0, 2/\{(1+r^2)L\}\right)$. $\mathrm{Id}\otimes\Sigma + \Sigma\otimes\mathrm{Id} - \gamma(\Sigma^{\otimes 2} + \mathrm{T})$ is thus invertible, which concludes the proof. $\qquad\square$

The covariance matrices make clearly visible the different origins of the noise. The Gaussian noise is responsible of the term $2\,\mathrm{Id}$, while the multiplicative and additive parts of the stochastic gradient (see (6)) are related to the operator $\mathrm{T}$ and to the term

$$\frac{\gamma N}{p}\sum_{i=1}^{N}\left(\frac{(x_i^{\mathrm{T}}\theta^\star - y_i)x_i}{\sigma_y^2} + \frac{\theta^\star}{\sigma_\theta^2}\right)^{\otimes 2} \tag{S22}$$

Figure S1: Illustration of Proposition 5, Theorem 6 and Theorem 7 in the asymptotic $N \to +\infty$. $\bar{\theta}$, $\bar{\theta}_{\mathrm{SGD}}$, $\bar{\theta}_{\mathrm{LMC}}$, $\bar{\theta}_{\mathrm{FP}}$ and $\bar{\theta}_{\mathrm{SGLD}}$ are the means under the stationary distributions $\pi$, $\pi_{\mathrm{SGD}}$, $\pi_{\mathrm{LMC}}$, $\pi_{\mathrm{FP}}$ and $\pi_{\mathrm{SGLD}}$, respectively. The associated circles indicate the order of magnitude of the covariance matrix. While LMC and SGLDFP concentrate to the posterior mean $\bar{\theta}$ with a covariance matrix of the order $1/N$, SGLD and SGD are at a distance of order $\sim 1$ of $\bar{\theta}$ and do not concentrate as $N \to +\infty$.

respectively.

Denote by

$$\eta_1 = \inf_{N \geq 1} \left\{ \frac{2N}{L} \wedge \frac{2N}{(1+r^2)L} \right\} > 0 . \tag{S23}$$

**Corollary 5.** *Consider the case of the Bayesian linear regression. Set $\gamma = \eta/N$ with $\eta \in (0, \eta_1)$ and assume that $\liminf_{N \to +\infty} N^{-1}m > 0$.*

$$\int_{\mathbb{R}^d} \|\theta - \theta^\star\|^2 \, \pi_{\mathrm{LMC}}(\mathrm{d}\theta) = d\Theta_{N \to +\infty}(N^{-1}) , \quad \int_{\mathbb{R}^d} \|\theta - \theta^\star\|^2 \, \pi_{\mathrm{FP}}(\mathrm{d}\theta) = d\Theta_{N \to +\infty}(N^{-1}) ,$$

$$\int_{\mathbb{R}^d} \|\theta - \theta^\star\|^2 \, \pi_{\mathrm{SGLD}}(\mathrm{d}\theta) = \eta d\Theta_{N \to +\infty}(1) , \quad \int_{\mathbb{R}^d} \|\theta - \theta^\star\|^2 \, \pi_{\mathrm{SGD}}(\mathrm{d}\theta) = \eta d\Theta_{N \to +\infty}(1) .$$

Recall that, according to the Bernstein-von Mises theorem, the variance of $\pi$ is of the order $d/N$ when $N$ is large. The corollary confirms that $\pi_{\mathrm{SGLD}}$ is very far from $\pi$ when the constant step size $\gamma$ is chosen proportional to $1/N$.

# 4 Illustration of Proposition 5, Theorem 6 and Theorem 7

We provide in Figure S1 an illustration of the results of Section 3.2 as the number of data items $N$ goes to infinity.