[Reviews · NeurIPS 2018]

Reviewer 1



Review after rebuttal: I thank the author(s) for their response. While I still believe that this paper is a minor increment beyond what has already been done on SGLD, I agree that the message might be useful for some. I also appreciate the effort the authors have made in improving the manuscript based on reviews' suggestions, particularly their efforts to include relevant numerical experiments to ML scenarios, and recommendations beyond the CV approach which has been studied to exhaustion and rarely applicable in practice. Based on this, I've adjusted my decision to marginally above threshold. Original review: In the paper "The promises and pitfalls of Stochastic Gradient Langevin Dynamics" the authors revisit the Stochastic Langevin Gradient Dynamics (SGLD) approach to approximately sampling from a probability distribution using stochastic gradients (specifically subsampling). The authors compare a number of different classes of approximate inference method, including SGLD, LMC (known by some as Unadjusted Langevin Algorithm or ULA) and Stochastic Gradient Langevin Dynamics Fixed Point (SGLDFP) -- the latter being a variant of SGLD with a control variate exploiting the unimodality of the distribution, similar to what has been presented in [3, 25 and others]. In the usual context of strong log-concave models, geometric convergence can be readily shown for all these algorithms via synchronous coupling arguments (Lemma 1) The authors investigate the asymptotic and non-asymptotic behaviour of these processes in the big-data setting (N = data size), under the frequently adopted assumption that the step size is proportional to 1/N and the mini-batch size is fixed independently of N. T They show that while the invariant measures for LMC and SGLDFP will converge to each other and the target distribution in Wasserstein-2 with rate O(N^{-1/2}), SGLD and SGD will converge to each other O(N^{-1/2}). They demonstrate an example (Bayesian linear regression) in which the distance to the invariant measure remains strictly bounded above from 0 as N -> infinity. To further understand the behaviour, the authors then obtain asymptotic formula for the first two moments of each respective invariant distribution -- this makes evident the bias with respect to the correct moments for SGD and SGLD. Finally, toy numerical experiments demonstrate the theory they have just described. Firstly, I commend the authors for being so precise. I found to be very clearly written. In my opinion, the main novelty in this paper is the demonstration that the equilibrium behaviour of (1) SGLD and SGD are so similar as N -> infinity and (2) LMC/ULA approaches the correct distribution as N->\infty (albeit this could have already been seen from other works). While the analysis is illuminating, the scope of applicability is quite limited, the assumption of strongly log-concave models does not put us too far from the Gaussian case, and has already been studied very well following the familiar lines of [A. Durmus and E.Moulines. High-dimensional Bayesian inference via the Unadjusted Langevin] [Dalalyan, Arnak S., and Avetik G. Karagulyan. "User-friendly guarantees for the Langevin Monte Carlo with inaccurate gradient." arXiv preprint arXiv:1710.00095 (2017).] [Dalalyan, Arnak S. "Further and stronger analogy between sampling and optimization: Langevin Monte Carlo and gradient descent." arXiv preprint arXiv:1704.04752 (2017).] and various other papers. Indeed, the proofs in the supplementary material follow almost identically similar proofs in these papers. The conclusion of this paper is that SGLD diverges from the correct distribution in the large N limit and to use control variates with SGLD. While the precise results differ, very similar observations using different approaches have been made in other papers, as the authors have mentioned themselves. My strongest criticism is that I feel that other avenues could have been addressed beyond introducing control variates which is now well established and which are unlikely to work beyond unimodal distributions, and thus strongly limited their usage in the ML community. For example, would another scaling of step size with N have fixed this asymptotic bias, or perhaps scaling the mini-batch size p with N -- indeed it appears that perhaps choosing p \propto N^{alpha} for some alpha > 0 would also be a solution? A negative or positive statement along these lines could have been easily considered in this work and provided additional useful recommendations to a practitioner. Moreover, given the typical style of NIPs papers I also believe the numerical experiments would have provided an opportunity to demonstrate the theory presented in a more realistic context outside the assumptions given. Perhaps demonstrating examples where the strong log-concavity assumption does not hold, or the model has multiple modes. In conclusion, while I believe this paper is clearly and precisely written, given the plethora of previous works investigating very similar ideas using very similar methods, and given the limited applicability of the conclusions for typical problems in ML, I would recommend that perhaps this work is not appropriate for NIPs in its current form, but would strongly recommend it be submitted to a more appropriate journal/conference. Finally, as a general comment that has no bearing on my decision on this paper, I find it curious that the authors claim that "SGLD has demonstrated spectacular successes in machine learning tasks". I am curious evidence this is based on -- I stand to be corrected but I have yet to see SGLD being used or applied in practice.

Reviewer 2



The authors consider a mainly theoretical investigation of the very widely used stochastic gradient Langevin dynamics. It has previously been shown that SGLD converges weakly to the target with decreasing step size, and in this paper the authors consider the more practically realistic setting of using a fixed stepsize. In particular they show that the stationary distribution of SGLD can be far from the true distribution as the number of data points N increases, using a fixed step size. In contrast, the SGLDFP algorithm (using control variates to reduce the variance of the gradient estimates) targets a distribution that gets closer to the true distribution as N increases. In my opinion this is a very useful addition to the theoretical literature in this area, as it justifies the use of SGLDFP with a constant step size, which is how it is usually employed in practice. Indeed I think papers of this type should be encouraged, where theory is driven by practical reality and subsequently influences utility. The paper, although rather technical and densely written, is quite readable and clearly defines the problem and setting being considered. The mathematics appears to be, to the best of my knowledge, correct. The numerical experiments section is very short, although I appreciate the difficulty of fitting such a dense paper into 8 pages. As a suggestion, figures 1 and 2 could be combined to free up some space for a short conclusions section, as the paper currently ends rather abruptly. A few very minor typos: Line 188: “On the opposite,” -> “In contrast,” Line 204: Remove “.” Line 87: “a MCMC” -> “an MCMC” References: Numerous typos and capitalisations need fixed, with line overruns in [6] and [27]

Reviewer 3



Rebuttal response: I appreciate the authors' thoughtful response. I think this is a timely paper that could be fairly impactful, particularly the SGD/LMC result. The strong log-concavity assumption does not bother me very much. Adding a Bayesian matrix factorization experiment to highlight greater ML relevance would be a welcome addition and substantially improve the paper. I've increased my score from 4 to 5. This issues with Section 3.2 prevent me from increasing it to 6. ---- Summary: The paper focuses on the theoretical properties of stochastic gradient Langevin dynamics (SGLD) and related algorithms in the constant step size regime. The author(s) show that in the large-data limit, constant step size SGLD behaves like stochastic gradient descent while SGLD with control variates (SGLDFD) behaves like the unadjusted Langevin algorithm. The large-data behavior of the means and covariances of the stationary distributions of SGLD, SGLDFD, SGD, and the unadjusted Langevin algorithm are also provided. Some simple numerical experiments with Bayesian logistic regression support the theory. I think the main contributions in Section 3.1 (described more below) are very illuminating on a number of levels. - First, it is nice to have non-asymptotic 2-Wasserstein bounds for SGLD and related methods available. To the best of my knowledge, previous work only gave 1-Wasserstein bounds. The fact the fact that SGLD with constant step size behaves like SGD nicely demonstrates its shortcomings. - Showing that SGLDFD and the unadjusted Langevin algorithm have similar stationary distributions illustrates that SGLDFD is a non-trivial improvement over SGLD. - Finally, the lower bound on the 2-Wasserstein distance between SGLD and SGLDFP shows that the upper bounds are reasonably tight. While all of these contributions are worthwhile, the paper is in need of revision to improve clarity. In particular, I was unsure what I was supposed to take away from Section 3.2. Thus, while I think the technical and scientific contributions are of NIPS quality, the presentation needs to be improved. Major comments: 1. In Section 3.1, it would be useful to add some discussion of what these bounds in Theorem 2 imply about the Wasserstein distance between SGLDFP and the exact posterior. This bound could then be contrasted with what is known about ULA. What choices of parameters would lead to a posterior approximation as good as ULA but at lower computational cost? 2. The Section 3.2 needs substantial revision. It introduces a lot of non-standard notation and reads like a math dump, with little explanation or intuition. What are the roles of L, eta_0, H, and G? How should we interpret Theorems 6 and 7? Minor comments - Line 27: “Stochastic Gradient Langevin Dynamics” => “stochastic gradient Langevin dynamics” - When a reference is used as a subject or object, use \citet{} instead of \citep{} (e.g. ref 33 on line 92) - It would be useful to clarify before Lemma 1 or after the assumptions are introduced that under H1 and H2, it always holds that m \le L - Line 187+: “On the opposite” => “On the other hand” - Line 195: C used in the previous display but then C* define in the following display - Line 202: italic L already defined as a constant in line 114. Choose a different letter or a more distinct font for the line 202 L - Line 202: the dot notation used to denote the application of the L operator is confusing. Just used standard function notation instead (e.g. “L \cdot A” => “L(A)”) Line 204: trailing period should be part of the previous display - Line 211: in the displays that follow, “C” => “C*” - Make sure capitalization is correct in bibliography. E.g., “mcmc” should be capitalized in ref. 23 - No need to include URLs in bib entries - arXiv preprints should include the arXiv ID